# S100A8/A9 predicts response to PIM kinase and PD-1/PD-L1 inhibition in triple-negative breast cancer mouse models
Lauren R. Begg[1], Adrienne M. Orriols [1,8], Markella Zannikou[1], Chen Yeh[1,2,9], Pranathi Vadlamani [1], Deepak Kanojia[1,10], Rosemary Bolin[3,11], Sara F. Dunne[4], Sanjeev Balakrishnan[5,12], Roman Camarda[5,13], Diane Roth[1], Nicolette A. Zielinski-Mozny[1,3], Christina Yau [5], Athanassios Vassilopoulos[1,6,14], Tzu-Hsuan Huang [1], Kwang-Youn A. Kim [1,2] & Dai Horiuchi [1,6,7] ✉

## Abstract

**Background** Understanding why some triple-negative breast cancer (TNBC) patients respond poorly to existing therapies while others respond well remains a challenge. This study aims to understand the potential underlying mechanisms distinguishing early-stage TNBC tumors that respond to clinical intervention from non-responders, as well as to identify clinically viable therapeutic strategies, specifically for TNBC patients who may not benefit from existing therapies.

**Methods** We conducted retrospective bioinformatics analysis of historical gene expression datasets to identify a group of genes whose expression levels in early-stage tumors predict poor clinical outcomes in TNBC. In vitro small-molecule screening, genetic manipulation, and drug treatment in syngeneic mouse models of TNBC were utilized to investigate potential therapeutic strategies and elucidate mechanisms of drug action.

**Results** Our bioinformatics analysis reveals a robust association between increased expression of immunosuppressive cytokine S100A8/A9 in early-stage tumors and subsequent disease progression in TNBC. A targeted small-molecule screen identifies PIM kinase inhibitors as capable of decreasing S100A8/A9 expression in multiple cell types, including TNBC and immunosuppressive myeloid cells. Combining PIM inhibition and immune checkpoint blockade induces significant antitumor responses, especially in otherwise resistant S100A8/A9-high PD-1/PD-L1-positive tumors. Notably, serum S100A8/A9 levels mirror those of tumor S100A8/A9 in a syngeneic mouse model of TNBC.

**Conclusions** Our data propose S100A8/A9 as a potential predictive and pharmacodynamic biomarker in clinical trials evaluating combination therapy targeting PIM and immune checkpoints in TNBC. This work encourages the development of S100A8/A9-based liquid biopsy tests for treatment guidance.

## Plain Language Summary

Breast cancer is a complex disease, and not all patients respond well to existing treatments. In this study, we sought to understand why some patients with a specific type of breast cancer called triple-negative breast cancer respond poorly to current therapies. We also aimed to identify new treatments for these patients. We analyzed genetic data from breast cancer patients and identified a factor called S100A8/A9, which is linked to poor outcomes in early-stage cancer. We tested drugs that can reduce the levels of this factor in tumors and found promising results, especially when combined with another treatment called immunotherapy. Our findings suggest that S100A8/A9 could help predict how patients will respond to treatments, potentially leading to better therapies in the future.

[1]Northwestern University Feinberg School of Medicine, Chicago, IL, USA. [2]Biostatistics Collaboration Center, Northwestern University, Chicago, IL, USA. [3]Center for Comparative Medicine, Northwestern University, Chicago, IL, USA. [4]High Throughput Analysis Laboratory, Northwestern University, Evanston, IL, USA. [5]University of California, San Francisco, San Francisco, CA, USA. [6]Robert H. Lurie Comprehensive Cancer Center, Northwestern University, Chicago, IL, USA. [7]Center for Human Immunobiology, Northwestern University, Chicago, IL, USA. [8]Present address: University of Florida College of Medicine, Gainesville, FL, USA. [9]Present address: Rush University Medical Center, Chicago, IL, USA. [10]Present address: Mythic Therapeutics, Waltham, MA, USA. [11]Present address: Pennington Biomedical Research Center, Baton Rouge, LA, USA. [12]Present address: Pulze.ai, San Francisco, CA, USA. [13]Present address: Novo Ventures US, Inc., San Francisco, CA, USA. [14]Present address: AbbVie, Inc., North Chicago, IL, USA. ✉e-mail: dai.horiuchi@northwestern.edu

TNBC, identified as lacking human epidermal growth factor receptor 2 (HER2) overexpression and functional expression of estrogen and progesterone receptors, is the breast cancer (BC) subtype with the poorest clinical outcome[1,2]. Molecular profiling efforts to understand the biology of TNBC in the past decade have uncovered as many as seven TNBC subclasses[3–5], revealing high levels of heterogeneity. These individual subclasses, identified based on gene expression patterns and genomic changes, all exhibit different degrees of association with clinical outcomes[3,6]. These molecular analyses—though highly informative—have not provided pivotal insight into why some TNBC patients respond poorly to existing therapies while others respond well.

S100A8 and -A9 belong to the S100 family of calcium-binding proteins that consists of 21 total members. S100A8 and -A9 are the only proteins in this family that must exist in a heterodimeric form (S100A8/A9) to maintain protein stability and function[7]. S100A8/A9 is overexpressed in several solid cancer types and has recently become recognized as a potential anticancer target[7–10]. While S100A8/A9 is classically shown to be involved in maintaining intracellular calcium homeostasis, it also functions as an immunosuppressive extracellular cytokine. S100A8/A9 is produced and secreted by multiple cell types in the tumor microenvironment (TME), including cancer cells and immune cells of the myeloid lineage, such as neutrophils, myeloid-derived suppressor cells (MDSCs), and tumor-associated macrophages (TAMs)[7,11–13].

S100A8/A9 is thought to support tumor progression via multiple mechanisms. Evidence suggests that potential mechanisms include the stimulation of cancer cell proliferation, and the accumulation of MDSCs and TAMs through autocrine and paracrine signaling[7,12–15]. These mechanisms could be initiated by the binding of S100A8/A9 to the growing list of prospective receptors, such as TLR4, RAGE, and IL-10R[7]. In preclinical studies, xenografted mouse cancer cells of several types grew slower with reduced metastatic burden in S100A9 knock-out mice[12,14,16]. Further, experimental anti-S100A8 and -A9 neutralizing antibodies reduced tumor growth and metastasis of a lung cancer model[8]. Intriguingly, it was recently reported that elevated S100A8/A9 expression in the TME was correlated with resistance to PD-1 blockade in melanoma patients[17]. Despite these findings, there remains lack of biomarker-directed therapeutic strategies in clinical development that consider the functionality and detectability of S100A8/A9.

The PIM family of serine/threonine kinases consists of three closely related members PIM1, −2, and −3[18] (PIM, hereafter). In clinical TNBC samples, elevated PIM expression was associated with poor outcomes[19,20]. PIM controls the activities of several target proteins via phosphorylation, including oncogenic transcription factor MYC[21], the pro-apoptotic BCL2 family member BAD[22], and the de facto tumor suppressor 4EBP1[23]. Notably, although PIM has not been well-researched in the context of the TME, an advanced single-cell RNA-sequencing analysis recently identified PIM in highly immunosuppressive MDSCs in a BC mouse model[24]. Additionally, bone-marrow-derived immunosuppressive mouse myeloid cells were found to depend on PIM for maintaining fatty acid metabolism and suppressive activity in vitro[25]. These findings suggest that PIM may promote disease progression through its well-documented roles in cancer cells, as well as its yet-to-be-established roles in other pro-tumorigenic cell types in the TME, such as immunosuppressive myeloid cells.

PIM is a constitutively active kinase with a structurally distinct ATP-binding pocket[26], enabling the development of selective small molecule inhibitors. In the past decade, four PIM kinase inhibitors have entered clinical studies, from the 1st-generation inhibitor SGI-1776[27] to the 2nd- and 3rd-generation inhibitors AZD1208[28], PIM447[29], and INCB053914[30]. Additionally, next-generation PIM inhibitors, such as GDC-0339[31], are currently under preclinical investigation. These inhibitors target multiple or all PIM family members and possess various extents of PIM-selectivity to counter functional compensation that occurs among them[32,33]. Emerging clinical data suggest that although a newer-generation inhibitor such as PIM447 can induce durable responses in some multiple myeloma patients, the efficacy of single-agent PIM inhibitor treatment is limited[34,35]. However,

no trials evaluating PIM inhibitors have employed a biomarker-informed patient stratification strategy.

This report identifies elevated S100A8/A9 expression in early-stage TNBC tumors as a strong predictor of subsequent disease progression, suggesting that its abundance, particularly in patients' peripheral blood samples, can serve as a readily usable prognostic biomarker. Using syngeneic mouse models of TNBC, this study also presents PIM kinase inhibition as a clinically feasible method to reduce the abundance of S100A8/A9 in the TME, making otherwise resistant S100A8/A9-high PD-1/PD-L1-positive tumors susceptible to immune checkpoint inhibitors (ICIs).

## Methods

### Bioinformatics analyses

The three publicly available gene expression datasets used to generate the data shown in Fig. 1 are identified as the following in Supplementary Data 1: GSE25066[36], Yau[37] (also identified in the literature as the chemotherapy-naïve historical dataset) available on the University of California, Santa Cruz (UCSC) Cancer Browser or UCSC Xena[38], and Chin[39]. Additionally, the TCGA breast invasive carcinoma dataset[40], available on the UCSC Cancer Browser or UCSC Xena, was used to perform additional confirmatory differential expression analysis.

The flow of the bioinformatics methods, schematically described in Supplementary Fig. 1a, is as follows: Samples were first assigned to one of two groups – low or high-risk, based on a method similar to that introduced by Bair and Tibshirani (2004)[41] (i.e., semi-supervised methods). For each dataset, gene-wise univariate Cox proportional hazards regression analysis was performed, and a score was calculated for the correlation of $\log_2$ expression of the respective genes to recurrence-free, progression-free, or distant-metastasis-free survival. Specifically, 5000 bootstrapped sample sets were generated for each dataset, and a Cox model was fit to each one. Following this, gene-wise bootstrap scores were obtained by trimming the top and bottom 5% of p-values based on the log-rank test and averaging the inverse of the remaining values[42]. Genes were then ranked based on these scores, and the top 300 candidates were used to develop a multi-gene model for sample stratification (Supplementary Data 1). Combinations of these candidates, comprised of genes added one at a time with respect to the rank order, were tested for their respective capacity to optimally stratify the samples. For each combination, k-means clustering was used to dichotomize the samples based on the expression values of the corresponding genes, and the following two metrics were calculated: the log-rank p-value of the difference in survivability of the two resulting groups and the cross-validation error rate of a nearest shrunken centroids classifier trained on this group classification. The latter was obtained using the pamr R package. The smallest list of genes that minimized these metrics was identified as the optimal multi-gene classifier. Group annotations for each dataset, obtained by applying the corresponding optimal classifier, were used for downstream differential gene expression analysis. Differential gene expression analysis was performed for each dataset using the limma R package. Genes significantly dysregulated in the high-risk group, relative to the low-risk samples, were obtained at a holm p-value cutoff of 0.05. The subset of these genes that are dysregulated in the same direction in at least 2 of the 3 clinical datasets queried was subsequently identified. To ensure that the size of this common subset of dysregulated genes is larger than what would be observed by chance alone, an empirical distribution of subset size was generated using gene-name sampling over 100,000 iterations, and the resulting one-tailed p-value of the observed subset size was calculated. These identified genes (Fig. 1b and Supplementary Data 1), when up- or down-regulated, can be associated with either poor or better outcomes.

For differential gene expression analysis shown in Supplementary Fig. 1d, log-transformed and median-centered S100A8 and -A9 expression values were derived for HER2 + , ER/PR + , and TNBC groups and visualized using the ggplot2 R package.

Kaplan-Meier analysis shown in Supplementary Fig. 1b, c is based on recurrence-free or distant metastasis-free survival. The optimal threshold used was identified by considering values between the 10th and 90th

percentile and selecting the cut-point that generated two groups with the most significant differences based on the log-rank test. The plots were generated using the *survival* R package.

Gene co-expression analysis was performed in cBioPortal[43] using the TCGA breast[40] and breast cancer METABRIC[44–46] datasets.

### Human and mouse TNBC cell lines and human mammary epithelial cells (HMECs)

A panel of established human TNBC cell lines, non-immortalized HMEC cells, and their culture conditions were previously described[47,48]. Non-immortalized HMEC cells were purchased from Lonza (CC-2551) at passage 5 and maintained in Mammary Epithelial Basal Medium, Phenol Red-free (Lonza, CC-3153), containing epidermal growth factor (5 ng/mL), hydrocortisone (500 ng/mL), insulin (5 µg/mL), bovine pituitary extract (70 µg/mL), transferrin (5 µg/mL), isoproterenol (10 µM), and GlutaMAX (Thermo Fisher, 35050061). The human cell lines described in Fig. 2a were utilized for conducting mechanistic experiments and obtaining genetically defined TNBC cell supernatants. It is important to note that the presence or absence of S100A8 and -A9, as shown in Fig. 2a, should not be interpreted as indicative of cancer cell sensitivity to commonly used chemotherapeutic agents.

The exogenous expression of PIM1 in MDA-MB-468 CRISPR-*PIM1,2,3* cells was achieved by infecting the cells with the lentivirus prepared using pLX_TRC317-*PIM1* (TRCN0000481389, Sigma), or pLX_TRC317-*empty* (kindly provided by Sigma), Lipofectamine 2000 transfection reagent (Thermo Fisher, 11668019), ViraPower Lentiviral Packaging Mix (Thermo Fisher, K497500), and Lenti-X 293 T cell line (Takara, 632180). Mouse D2A1-M1 and -M2 cell lines, generously provided by Drs. Clare Isacke and Ute Jungwirth, were previously described[49], and the Py8119 cell line was purchased from ATCC (CRL-3278). The Universal Mycoplasma Detection Kit (ATCC, 30-1012 K) was used to ensure that cells were not infected with mycoplasma.

### Small-molecule inhibitors and human S100A8/A9 ELISA screen

GDC-0339, used in the experiments described in Fig. 2, was purchased from MedChem Express (HY-16976). GDC-0339 used elsewhere, including all in vivo experiments, was provided by Genentech. The other 29 targeted anticancer agents used in this study were purchased from Selleck Chemicals and TargetMol.

For the human S100A8/A9 ELISA screen, MDA-MB-468 cells were seeded onto a 96-well plate at 5000 cells/100 µL media/well, incubated for 24 h, and subjected to drug treatment for 72 h as described in Fig. 2b. At the end of 72-h treatment, supernatants were collected, and S100A8/A9 concentrations were determined using the human S100A8/A9 heterodimer ELISA kits (R&D Systems, DS8900, and DY8226-05) according to the manufacturer's instructions. Colorimetric signals were acquired and processed using the Cytation 5 Cell Imaging Multi-Mode Reader (BioTek). A typical concentration of S100A8/A9 in control (DMSO)-treated well was 40-80 ng/mL. The drug treatment experiment was independently repeated three times.

### CRISPR experiments

CRISPR experiments were performed as previously described[48] using Gene Knockout Kits v2 (Synthego) for human *PIM1*, -*2*, -*3*, and *S100A8* (with each Kit containing a mixture of three target-specific, synthetic multi-guide sgRNA sequences and purified Cas9 nuclease) and Lipofectamine CRISPRMAX Cas9 Plus transfection reagent (Thermo Fisher, CMAX00008) according to the protocol (entitled "CRISPR editing of immortalized cell lines with RNPs using lipofection") provided by Synthego. Non-targeting sgRNAs (Negative control scrambling sgRNA #1 and #2) were also purchased from Synthego and mixed at 1:1 before use. sgRNA/Cas9 reverse-transfection was performed in 24-well plates (3.9 pmol sgRNA + 3 pmol Cas9 per well), and the resulting heterogeneous populations of cells were expanded in 6-well plates. The sgRNA/Cas9-treated cells were used in downstream assays within ~six weeks of transfection to avoid the expansion of undesired cell populations. Each biological replicate was initiated by sgRNA/Cas9 transfection. We found that while MDA-MB-468 cells were readily amenable to CRISPR-based genetic manipulation, BT-20 cells and freshly isolated mouse CD11b + Gr1+ myeloid cells were not, due to transfection-induced excessive toxicities.

### siRNA experiments

siRNA products were purchased from Thermo Fisher (Silencer Select siRNA). The specific products used were: *S100A8* #1, 2 (s12422. S12423), *S100A9* #1, 2 (s12425, s12426), *STAT3* #1, 2 (s744, s745), *CEBPB* #1, 2 (s2891, s2892), *MYC* #1, 2 (s9129, s9130), *PPARG* #1, 2 (s10887, s10888), Negative Control (#2, 439084), and *GAPDH* (7390850). Negative Control and *GAPDH* siRNA sequences were mixed at 1:1 and used as the sole non-specific control. siRNA transfection was performed using FuGENE SI Transfection Reagent (Fugent LLC, SI-1000) via reverse transfection. siRNAs were used at 60 pmol per gene per well in 6-well culture dishes, making the final concentration of siRNAs in each well approximately 50 nM. siRNA-transfected cells were harvested for western analysis 72 h after transfection.

### Western blotting and immunoprecipitation

Cultured cells were washed with ice-cold PBS and lysed in radio-immunoprecipitation assay (RIPA) buffer (50 mM Tris, 150 mM NaCl, 0.5% sodium-deoxycholate, 1% NP-40, 0.1% SDS, 2 mM EDTA, pH 7.5) containing protease inhibitor cocktail (Thermo Fisher, A32955) and phosphatase inhibitors (Thermo Fisher, A32957). Isolated tumor tissues were first rinsed in ice-cold PBS and homogenized on ice using a powered tissue homogenizer (OMNI Tissue Master 125) in RIPA buffer containing protease inhibitors and phosphatase inhibitors. Protein concentrations were determined using the DC Protein Assay (Bio-Rad, 5000112) with BSA as the standard. The ECL reaction was done using the Clarity Western ECL Substrates (Bio-Rad, 170561) or Radiance Plus Chemiluminescent Substrate (Azure Biosystems, AC2103), and chemiluminescent signals were acquired with the Bio-Rad ChemiDoc Touch Imaging system equipped with a supersensitive CCD camera. Where indicated, unsaturated band intensities were quantified using Bio-Rad Image Lab software (Version 6.1.0 build 7). Actin was used as the loading control and to quantify western bands. Throughout this study, western membranes were horizontally cut at specific molecular weights before incubation with previously characterized antibodies, enabling the simultaneous detection of more than one non-overlapping target. When it was not possible to simultaneously detect targets due to close proximity, the same cut membranes were treated with Restore Western Blot Stripping Buffer (Thermo Fisher, 21059) according to the manufacturer's instructions, and subsequently re-probed with a different antibody. The western blots in the figure panels represent conclusions from addressing underlying scientific questions in at least three separate experiments. For the immunoprecipitation experiment in Fig. 3f, approximately 750 µg of solubilized proteins were incubated with 1 µg of anti-C/EBPβ mouse IgG (Santa Cruz Biotech, sc-7962) or normal mouse IgG (Santa Cruz Biotech, sc-2025) at 4 degrees before the addition of equilibrated protein G agarose beads (Santa Cruz, sc-2002).

The antibodies used for western blotting in this study and their working concentrations are: βActin (clone C4, Santa Cruz Biotechnology, sc-47778 HRP, 1:10000), human PIM1 (clone ZP003, Abcam, ab54503, 1:500), mouse PIM1 (clone C93F2, Cell Signaling Technology, 3247, 1:750), human PIM2 (clone D1D2, Cell Signaling Technology, 4730, 1:500), mouse PIM2 (clone EPR6987, Abcam, ab129057, 1:10000), PIM3 (clone D17C9, Cell Signaling Technology, 4165, 1:500), human S100A8 (clone C-10, Santa Cruz Biotechnology, sc-48352 HRP, 1:500), mouse S100A8 (R&D Systems, AF3059, 1:500), human S100A9 (clone 6B4, Millipore Sigma, MABF854, 1;500), mouse S100A9 (clone 372510, R&D Systems, MAB2065, 1:500), STAT3 (clone 124H6, Cell Signaling Technology, 9139, 1:1000), STAT3 pS727 (Cell Signaling Technology, 9134, 1;1000), C/EBPβ (clone H-7, Santa Cruz Biotechnology, sc-7962 HRP, 1:200), C/EBPβ pT235 (Cell Signaling Technology, 3084, 1:1000), c-MYC (clone Y69, Abcam, ab32072, 1:2000),

PPARG (clone C26H12, Cell Signaling Technology, 2435, 1:1000), Ubiquitin (clone P4D1, Cell Signaling Technology, 14049, 1:1000), CD45 (clone EPR20033, Abcam, ab208022, 1:1000).

## Real-time quantitative PCR (qPCR)

Total RNA from the human cell lines and mouse tumor samples was extracted using RNeasy Plus Mini Kits (Qiagen, 74134) according to the manufacturer's instructions. Real-time PCR reactions were run on the CFX96 Touch Real-Time PCR Detection System (Bio-Rad) paired with the CFX Manager software (Bio-Rad) using TaqMan probes (Thermo Fisher) specific for the following genes: human *S100A8* (Hs00374264_g1), human *S100A9* (Hs00610058_m1), mouse *S100A8* (Mm00496696_g1), and mouse *S100A9* (Mm00656925_m1). Gene expression was normalized to the *GAPDH* housekeeping gene (Hs00266705_g1 for the human gene and Mm99999915_g1 for the mouse gene).

## In vivo efficacy experiments

The animal efficacy experiments described in this study were approved by the Northwestern University Institutional Animal Care and Use Committee (Protocol ID: IS00002202) and were executed in collaboration with the Preclinical Rodent Services team at the Center for Comparative Medicine (CCM) of Northwestern University, which did not possess the conceptual understanding of the experimental groups. BALB/cJ and C57BL/6 J mice used in this study were acquired from The Jackson Laboratory (#000651 and #000664, respectively) and housed in pathogen-free rodent holding rooms within the CCM, where room temperature was approximately 22 °C. Throughout this study, mice were maintained in ventilated micro-isolator cages connected to an automatic water system, fed standard non-high-fat chow, and subjected to a 12-hour light/dark cycle. All syngeneic mouse cancer cell lines-D2A1-M1, -M2, and Py8119-were tested by Charles River Research Animal Diagnostics Services for infectious agents before in vivo use.

To grow syngeneic tumors, the cells ($5 \times 10^5$ cells for D2A1-M1 and -M2, and $1 \times 10^6$ cells for Py8119), resuspended in PBS, were percutaneously injected into the fourth mammary glands of female mice (BALB/cJ for D2A1-M1, and -M2, and C57BL/6 J for Py8119) aged 10-12 weeks. For efficacy experiments, the tumors were allowed to reach approximately 75-100 mm³ in volume, at which time drug treatment was initiated. Tumor-bearing animals were treated with the following agents: GDC-0339, a pan-PIM kinase inhibitor obtained from Genentech, reconstituted in 0.5% w/v methylcellulose/0.15% Tween 80 in in-vivo grade water, according to the reconstitution protocol kindly provided by Genentech; anti-mouse PD-1 antibody (BioXCell, clone RMP1-14, BE0146), anti-mouse PD-L1 antibody (BioXCell, clone 10 F.9G2, BE0101), and anti-mouse CD8β antibody (BioXCell, clone 53-5.8, BE0223), all diluted in InVivoPure pH 7.0 Dilution Buffer (BioXCell, IP0070) before use. GDC-0339 was administered via oral gavage, and the antibodies were administered via intraperitoneal injection.

## Determination of S100A8/A9 concentrations in mouse and human serum samples

Serum S100A8/A9 levels were determined using the Mouse S100A8/A9 Heterodimer DuoSet ELISA kit (R&D Systems, DY8596-05) and the Human S100A8/A9 heterodimer ELISA kit (R&D Systems, DY8226-05), in combination with ChonBlock Blocking/Sample Dilution ELISA Buffer (Chondrex, 9068) and ChonBlock Detection Antibody Dilution Buffer (Chondrex, 90681) designed to minimize serum-derived background signals. Mouse serum samples were isolated from whole blood collected from mice via cardiac puncture. Human serum samples from healthy donors and patients with TNBC (Fig. 5f) were acquired from BioIVT, LLC.

## Immune cell identification in mouse tumor and spleen samples

Tumor samples harvested from mouse mammary glands were dissociated using the Mouse Tumor Dissociation Kit (Miltenyi Biotec,130-096-730), gentleMACS C tubes (Miltenyi Biotec, 130-093-237), and gentleMACS Octo Dissociator with heaters (Miltenyi Biotec) following manufacturer

protocols. Where indicated, the resulting single cells were sorted on CD45 using EasySep™ Mouse CD45 Positive Selection Kit (StemCell Technologies, 18945) and used for downstream assays. Spleens were harvested from mice and mechanically dissociated between the frosted ends of two glass slides to create a cell suspension. After washing with PBS containing 2% FBS and lysis of red blood cells with ACK lysing buffer (Gibco, A1049201), resulting single cells were resuspended in PBS containing 2% FBS, and cell viability was determined using Muse Cell Analyzer (Luminex) before staining with flow antibodies. The following markers were used to identify specific immune cell types. CD8 T cells: CD45 + CD3 + CD8 + CD4-, Tregs: CD45 + CD3 + CD4 + CD8-FoxP3 + , and MDSCs: CD45 + CD11b + CD84+ (Ly6C + Ly6G-)/(Ly6C-/Ly6G + ).

The antibodies used for flow staining in this study are as follows, and diluted at 1:100 unless otherwise indicated: rat anti-mouse/human CD11b conjugated to BV605 (clone M1/70, Biolegend, 101257), hamster anti-mouse CD3e conjugated to BV510 (clone 145-2C11, BD Biosciences, 563024), rat anti-mouse CD4 conjugated to APC-Cy7 (clone GK1.5, Biolegend, 100414), rat anti-mouse CD45 conjugated to BUV395 (clone 30-F11, BD Biosciences, 564279), rat anti-mouse CD45 conjugated to PE-Cy7 (clone 30-F11, Biolegend, 103113), Armenian hamster anti-mouse CD84 conjugated to PE (clone mCD84.7, Biolegend, 122806), rat anti-mouse CD8a conjugated to BUV805 (clone 53-6.7, BD Biosciences, 612898), rat anti-mouse FOXP3 conjugated to PE (diluted 1:20, clone MF-14, Biolegend, 126404), eBioscience Fixable Viability Dye eFluor 450 (Thermo Fisher, 65-0863-14), eBioscience Fixable Viability Dye eFluor 520 (Thermo Fisher, 65-0867-14), rat anti-mouse Ly6C conjugated to APC-Cy7 (clone HK1.4, Biolegend, 128026), rat anti-mouse Ly6C conjugated to BV421 (clone HK1.4, Biolegend, 128031), rat anti-mouse Ly6G conjugated to PerCP-Cy5.5 (clone 1A8, BD Biosciences, 560602), rat anti-mouse PD-1 (CD279) conjugated to BV421 (clone RMP1-30, Biolegend, 109121), rat anti-mouse PD-L1 (CD274) conjugated to APC (clone 10 F.9G2, Biolegend, 124311).

## In vitro drug treatment of isolated mouse CD11b + Gr1+ myeloid cells

CD11b + Gr1+ myeloid cells were isolated from treatment-naïve D2A1-M1 tumors at approximately 1000mm³ in volume, using EasySep™ Mouse MDSC (CD11b + Gr1 + ) Isolation kit (StemCell Technologies, 19867) according to the manufacturer's instructions. Contaminating cancer cells, which rapidly form cell aggregates in vitro, were removed via gravitational sedimentation (10 min at 37 degrees) in a CO2 incubator immediately following CD11b + Gr1+ cell isolation. Although this isolation kit is marketed as an MDSC isolation kit, the isolated cell population is indicated as CD11b + Gr1+ myeloid cells in this report (e.g., Fig. 4d), considering the recent identification of CD84 as a new marker that distinguishes MDSCs from neutrophils[24]. Isolated CD11b + Gr1+ cells were plated onto a 24-well ultra-low attachment plate ($1 \times 10^6$ cells/well) (Corning, 3473) in RPMI supplemented with 10% FBS, 1% penicillin/streptomycin (Gibco 35050-061), 20 ng/ml murine GM-CSF (PeproTech 315-03), and 30% 4T1-conditioned RPMI, prepared as previously described[50]. Plated cells were then treated with pan-PIM inhibitors. The use of ultra-low attachment plates was intended to prevent the isolated CD11b + Gr1+ cells from adhering to the plates, thereby losing the characteristics of myeloid cells.

## Chemotaxis, cell proliferation, and activity assays using human PBMC-derived cells

CD11b+ cells used in the chemotaxis assay described in Fig. 5g were isolated from human peripheral blood mononuclear cells (PBMCs) (StemCell Technologies, 70025, individual donors selected) using EasySep™ Human CD11b Positive Selection and Depletion Kit (StemCell Technologies, 100-0742) according to the manufacturer's instructions. Isolated cells were resuspended in RPMI 1640 media supplemented with 10% FBS, 1x GlutaMax (Gibco, 35050-061), 1x MEM Non-Essential Amino Acid Solution (Gibco, 11140-050), 1x Sodium Pyruvate (Gibco, 11360-070), and 1×2-mercaptoethanol (Gibco, 21985-023). Transwell™ (5 µm pore) polycarbonate membrane inserts (Corning, 3421) containing approximately

600 K CD11b+ cells in 400 μL of complete media were loaded onto a 24-well ultra-low attachment plate (Corning, 3473) containing 550 μL/well of conditioned media for cell migration to occur. Conditioned media were prepared by culturing MDA-MB-468 CRISPR lines in 10 cm dishes (80-90% confluency) for 48 h in regular RPMI media (RPMI 1640 supplemented with 10% FBS). After 8 h of incubation, the cells found in the lower chambers were counted (Muse Cell Analyzer, Luminex) or processed for flow analysis. Although not commonly described, we found that small percentages of CD11b+ cells could migrate through 5 μm pores to reach the lower chamber in the absence of any potential chemoattractant (e.g., fresh RPMI) presumably via gravity, thereby creating unwanted noise in conducting cell counting. Thus, the numbers corresponding to non-specific backgrounds were subtracted when calculating the numbers of migrated cells attributable to conditioned media. Approximately 300 K cells correspond to 100 (%) in Fig. 5h.

For cell proliferation and activity analysis described in Supplementary Fig. 5b, human CD8 + T cells (StemCell Technologies, 70027), CD4 + T cells, isolated from PBMCs (StemCell Technologies, 70025) using EasySep™ Human CD4 + T Cell Isolation Kit (StemCell Technologies, 17952), and human CD11b+ cells (described above) were incubated with 50%/50% media (PBMC-compatible RPMI/conditioned media, described above). All cell types were plated at 100 K cells in 200μl 50%/50% media in U-bottom 96-well ultra-low attachment plates (Corning, 3474). CD8 + T cells were stained with CellTrace Violet Proliferation Kit (Thermo Fisher Scientific, C34557) per manufacturer instructions. CD8 + T cells and isolated CD4 + T cells were activated using Immunocult Human CD3/CD28/CD2 T cell activator (StemCell Technologies, 10970) per manufacturer instructions; T cell activator was not added to isolated CD11b+ cells. All cell types were cultured for 48 h. 5 h prior to harvesting/processing cells for flow analysis, eBioscience Protein Transport Inhibitor Cocktail (Thermo Fisher, 00-4980-93) was added to CD8+ and CD4 + T cells according to manufacturer instructions.

For all experiments utilizing downstream flow analysis, the following markers were used to identify specific cell types. CD8 T cells: CD45 + CD3 + CD8 + CD4-, Tregs: CD45 + CD3 + CD4 + CD8-FoxP3 + CD25 +, MDSCs: CD45 + CD11b + CD84 + (CD14 + HLA-DR$^{lo,neg}$CD33 + )/(CD15 + HLA-DR$^{lo,neg}$), Neutrophils: CD45 + CD11b + CD84-CD16 + . CD8 + T cells were fixed/permeabilized using Cytofix/Cytoperm Fixation/ Permeabilization kit (BD Biosciences 554714) per manufacturer instructions; CD4 + T cells were processed using True-Nuclear Transcription Buffer Set (Biolegend, 424401) per manufacturer instructions. The effects of conditioned media on proliferation and activity of different immune cell types are shown as relative changes, instead of absolute numbers, in Supplementary Fig. 5c. This is due to differences observed in the relative abundance and activation levels of different cell types in PBMCs between biological replicates.

The antibodies used for flow staining in this study are: mouse anti-human CD3 conjugated to BUV737 (clone UCHT1, BD Biosciences, 612750), mouse anti-human CD45 conjugated to FITC (clone HI30, Biolegend, 304005), mouse anti-human FOXP3 conjugated to BB700 (clone 236 A/E7, BD Biosciences, 566526), mouse anti-human CD69 conjugated to PE (clone FN50, Biolegend, 310905), mouse anti-human TGF-β conjugated to PE/Cyanine7 (clone S20006A, Biolegend, 300007), rat anti-human IL-10 conjugated to APC (clone JES3-19F1, BD Biosciences, 554707), rat anti-human CD4 conjugated to Alexa Fluor 700 (A161A1, Biolegend, 357417), eBioscience Fixable Viability Dye conjugated to eFluor 450 (Thermo Fisher Scientific, 65-0863-18), eBioscience Fixable Viability Dye conjugated to eFluor 520 (Thermo Fisher Scientific, 65-0867-18), mouse anti-human CD8 conjugated to BV510 (clone SK1, BD Biosciences, 563919), mouse anti-human CD25 conjugated to BV650 (clone M-A251, BD Biosciences, 563719), mouse anti-human CD11b conjugated to BV395 (clone ICRF44, BD Biosciences, 563839), mouse anti-human CD14 conjugated to PerCP/Cyanine5.5 (clone HCD14, Biolegend, 325621), mouse anti-human CD84 conjugated to PE (clone CD84.1.21, Biolegend, 326007), mouse anti-human CD15 conjugated to PE/Cyanine7 (clone HI98,

BD Biosciences, 560827), mouse anti-human CD33 conjugated to PE/Dazzle 594 (clone WM53, Biolegend, 303431), mouse anti-human CD16 conjugated to BV510 (clone 3G8, Biolegend, 302047), mouse anti-human HLA-DR conjugated to BV605 (clone L243, Biolegend, 307639), mouse anti-human CD45 conjugated to BUV395 (clone HI30, BD Biosciences, 563791), mouse anti-human/mouse Granzyme B conjugated to PerCP/Cyanine5.5 (clone QA16A02, Biolegend, 372211), mouse anti-human Perforin conjugated to PE/Cyanine7 (clone B-D48, Biolegend, 353315), mouse anti-human TNF-α conjugated to PE/Dazzle 594 (clone MAb11, Biolegend, 502945), mouse anti-human CD8 conjugated to APC (clone SK1, Biolegend, 344721), mouse anti-human IFN-γ conjugated to BV605 (clone B27, Biolegend, 506541). All antibodies were diluted at 1:20, using 5μL/test per manufacturer instructions.

**NanoString PanCancer IO360™ panel**
Total RNA samples were extracted from D2A1-M1 tumor samples collected at the end of the 2-week drug treatment experiment (Fig. 4e) using a powered tissue homogenizer (OMNI Tissue Master 125) and RNeasy Plus Mini Kits (Qiagen, 74134) according to the manufacturer's instructions. All the samples, with one exception (#24 in Supplementary Data 4), were prepared in nuclease-free water at 20 ng/mL (x 100 μL). #24 was prepared at 2 ng/mL (x 50 μL) due to low RNA yield. The concentration-adjusted RNA samples were sent to NanoString Technologies, Inc. (Seattle, WA) and analyzed on the PanCancer IO360™ gene expression panel using the NanoString nCounter platform. Data analysis, including quality control and normalization, was performed by NanoString, Inc., which did not possess a conceptual understanding of the experimental groups, using the nSolver and Rosalind analysis software. The figure panels presented in Fig. 4h and Supplementary Fig. 4e were generated by the NanoString scientists for fees using the raw data contained in Supplementary Data 4.

**Statistics and reproducibility**
Unless otherwise indicated, all results are shown as mean +/- SEM. Statistical analyses were performed using Prism 9 (version 9.5.0 build 525) from GraphPad Software, Inc. and R (Version 4.2.2), taking into consideration the assumptions required for the respective tests. $P < 0.05$ was considered to indicate statistical significance throughout the study. All cell-based in vitro experiments were independently repeated at least three times in duplicate or triplicate, depending on each assay. No statistical method was used to predetermine the sample size throughout this study. For animal experiments, efforts were made to achieve this study's scientific goals with the minimum number of animals. With respect to randomization, for animal experiments, tumor-bearing mice of similar tumor burden were equally divided into the control and experimental groups for subsequent drug treatment. No experimental samples were excluded throughout this study, except for animals that experienced unexpected, acute illness or injury, per the veterinarian's order.

**Reporting summary**
Further information on research design is available in the Nature Portfolio Reporting Summary linked to this article.

## Results
### Elevated S100A8/A9 expression in early-stage tumors is robustly associated with the risk of disease progression in TNBC
To understand potential mechanisms that differentiate curable or manageable TNBC tumors from those that still progress despite utilizing existing therapies, we employed a rigorous computational approach known as a semi-supervised method[41]. This gene expression analysis method differs markedly from those reported to date, such as the method described for identifying the seven TNBC subclasses, in that it considers gene expression data and patient prognostic data simultaneously to help reduce intrinsic arbitrariness (Supplementary Fig. 1a and Methods). We applied this method to three independent, publicly available datasets accompanied by well-annotated clinical information[36,37,39] to retrospectively identify genes whose

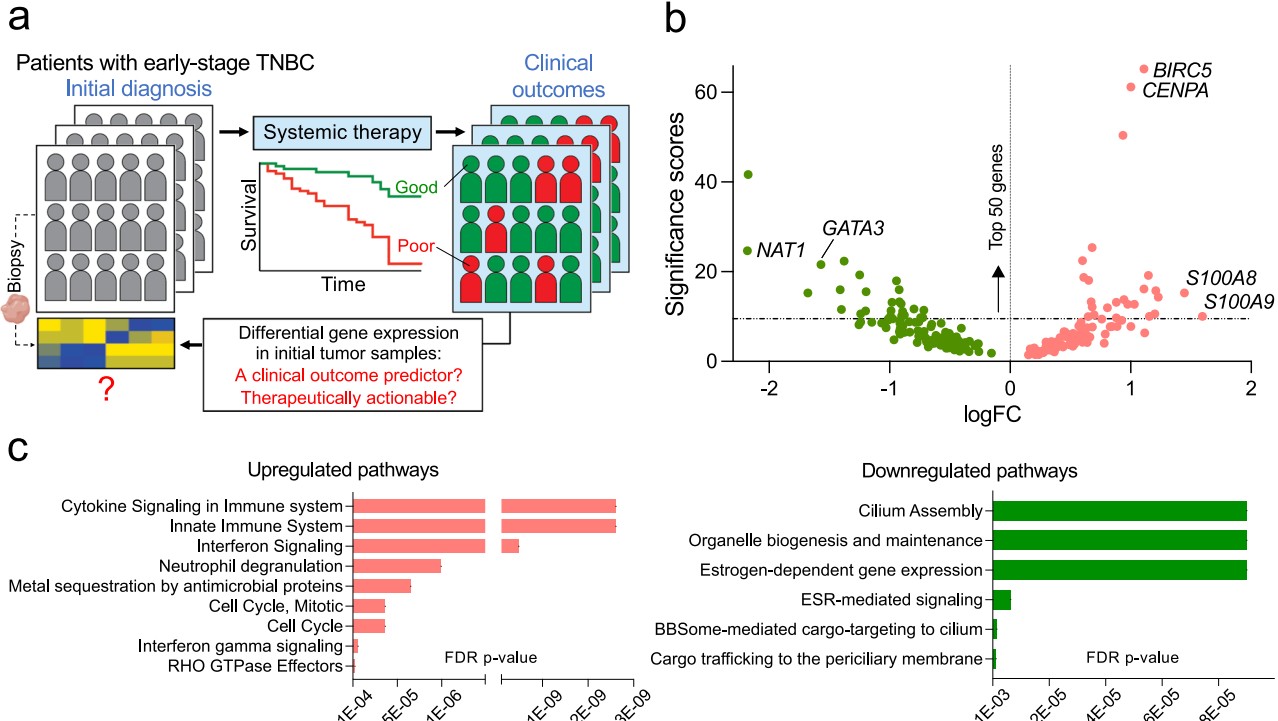

**Fig. 1 | Identification of poor clinical outcome-associated genes in TNBC.**
**a** Schematic representation of the overarching research questions addressed in this study. **b** A volcano plot showing the distribution of 210 genes whose expression levels in stage 1-3 tumors are most robustly associated with subsequent disease progression in TNBC. Significance scores are derived from the relative significance, as calculated by the methods described in Supplementary Fig. 1a and as shown in Supplementary Data 1 (scores in the furthest right column). FC: fold-change. **c** Biological pathways, based on the Reactome database, mediated by the upregulated or downregulated poor outcome-associated genes identified in our bioinformatics analysis (Fig. 1b and Supplementary Data 1). FDR: false discovery rate; FDR-adjusted *p*-values.

expression levels in stage 1-3 tumors were strongly associated with subsequent disease progression. Importantly, we selected these three cohorts as primary tumors were collected prior to beginning systemic therapies. This allowed us to identify a group of dysregulated genes in these tumors independent of confounding variables, such as biological changes triggered by systemic treatments (Fig. 1a).

In our analysis, we identified our genes of interest based on the criterion that they are dysregulated in the same direction (up- or down-regulated) in at least two of the three datasets used in this study. This analysis identified 210 genes, of which 99 were found to be upregulated (Fig. 1b and Supplementary Data 1). Among those upregulated genes, the most prognostic included cell division-related genes *BIRC5* and *CENPA*. *BIRC5*, encoding Survivin[51], a member of the inhibitor of apoptosis family, has been established as a predictor of poor outcomes in BC[52] (Fig. 1b and Supplementary Data 1). The most predictive genes among those downregulated included *NAT1*, encoding a drug-metabolizing enzyme N-acetyltransferase 1[53], and *GATA3*, encoding a transcription factor GATA Binding Protein 3 associated with epithelial cell terminal differentiation[54] (Fig. 1b and Supplementary Data 1). Increased expression of *NAT1* and *GATA3* has been independently associated with favorable prognosis in hormone-receptor-positive BC[55,56]. Although these genes have been poorly studied in TNBC, our data suggest they may mechanistically influence how TNBC responds to systemic treatments. Subjecting the 210 dysregulated genes to the Reactome pathway analysis[57] revealed that specific components of the immune system responses, including cytokine signaling, were most significantly overactive. In contrast, those genes constituting organelle biogenesis and hormone-dependent pathways were markedly downregulated, albeit less significantly (Fig. 1c and Supplementary Data 2).

Among the factors identified, we pursued S100A8/A9 for further studies. We chose S100A8/A9 because our bioinformatics approach ranked both *S100A8* and -*A9* within the top 50 out of the 210 dysregulated genes

identified in our analysis (Fig. 1b and Supplementary Data 1). Combined with the body of functional evidence available in the literature[12–14,16], this suggested that S100A8/A9 might tangibly contribute to poor outcomes, rather than being merely correlated with them, and thus could represent a potential therapeutic target. The validity of S100A8/A9 as a poor outcome predictor was further established by the Kaplan-Meier estimator, used as a confirmatory method (Supplementary Fig. 1b). Intriguingly, although we initially identified S100A8/A9 as a prognostic factor among TNBC patients, we also found that its prognostic significance persisted when all BC subtypes were analyzed, including the samples positive for HER2 and hormone receptors (Supplementary Fig. 1c). In agreement with this finding, receptor-status-specific differential gene expression analysis of *S100A8* and -*A9* in the datasets used in this study showed that *S100A8* and -*A9* levels were significantly and similarly elevated in TNBC samples as well as in HER2-positive samples when compared to hormone receptor-positive samples (Supplementary Fig. 1d). Therefore, these observations indicate that the prognostic significance of elevated S100A8/A9 expression is not determined exclusively by specific clinical BC subtypes (i.e., receptor status).

**PIM kinases control S100A8/A9 expression and secretion**
An accumulation of evidence suggests that S100A8/A9 can produce an immunosuppressive TME[7,14], potentially contributing to poor outcomes in cancer patients. However, despite S100A8/A9 being acknowledged as an antibody-druggable target, no such agents have been developed for clinical use. This may be partly due to a lack of consensus on the binding properties of S100A8/A9, specifically when and how it binds to specific cell types via prospective receptors. As expected, this may make the design of neutralizing antibodies challenging for use in human patients. Given this current reality, we instead sought to explore whether there might be existing clinically viable small molecules that could reduce S100A8/A9 expression or secretion. To this end, we utilized a human TNBC cell line, MDA-MB-468, one of the cell

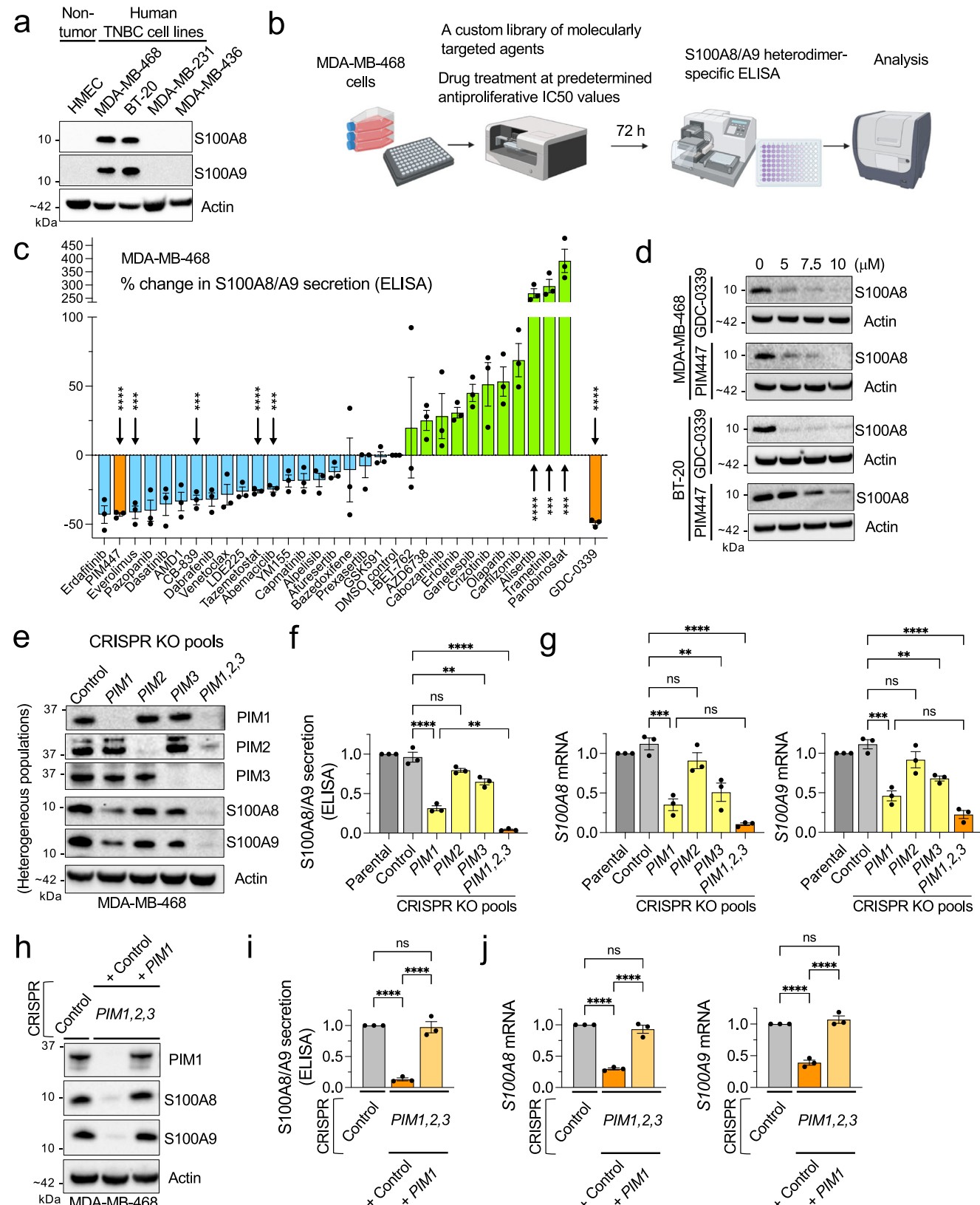

lines that we found to express S100A8/A9 proteins abundantly (Fig. 2a), and confirmed that S100A8 and -A9 in heterodimeric form is required to maintain protein stability within this model (Supplementary Fig. 2a). We then conducted a screen in which we evaluated 30 targeted anticancer agents for their ability to reduce S100A8/A9 secretion by MDA-MB-468. The 30

agents selected for our screen target a wide range of pro-tumor pathways, including metabolism, cell cycle, cell death, epigenetics, DNA repair, stress response, and protein synthesis/degradation (Supplementary Table 1). These agents are either FDA-approved or under active clinical evaluation and have been used as single-agents in preclinical BC models. Using an

**Fig. 2 | Identification of PIM kinases as factors controlling S100A8/A9 expression in TNBC cells. a** Representative western blots showing protein expression levels of S100A8 and -A9 in the indicated human TNBC cell lines and non-tumorigenic human mammary epithelial cells (HMEC). **b** Schematic representation of the ELISA screen performed in this study. MDA-MB-468 cells were treated with 30 targeted anticancer agents at their respective half-maximal inhibitory concentration (IC50) (Supplementary Table 1) or 10 μM, whichever was lower. **c** Results from the ELISA screen described in Fig. 2b ($n = 3$). **d** Representative western blots showing the effects of the indicated pan-PIM kinase inhibitors on S100A8 expression in the indicated TNBC cell lines (72 h). **e** Representative western blots showing expression of PIM1, -2, -3, S100A8, and -A9 in the cells transiently transfected with Cas9 proteins and the indicated *PIM*-specific, synthetic multi-guide sgRNA or a nonspecific control sgRNA. The resulting cells represent heterogenous KO pools and not single-cell clones. **f** Relative S100A8/A9 secretion (72 h) in Cas9/*PIM*-specific, sgRNA-

transfected MDA-MB-468 cells shown in Fig. 2e ($n = 3$). **g** Relative mRNA expression of *S100A8* (left) and -*A9* (right), as determined by qPCR, in Cas9/*PIM*-specific, sgRNA-transfected MDA-MB-468 cells shown in Fig. 2E ($n = 3$). **h** Representative western blots showing the effects of exogenously introduced PIM1 on S100A8 and -A9 expression in the cells transiently transfected with Cas9/*PIM1,2,3*-specific sgRNAs. **i** The effects of exogenously introduced PIM1 on relative S100A8/A9 secretion in MDA-MB-468 cells transiently transfected with Cas9/*PIM1,2,3*-specific sgRNAs ($n = 3$). **j** The effects of exogenously introduced PIM1 on relative mRNA expression of *S100A8* (left) and -*A9* (right), as determined by qPCR, in MDA-MB-468 cells transiently transfected with Cas9/*PIM1,2,3*-specific sgRNAs ($n = 3$). Throughout this figure, actin serves as a loading control. Error bars represent means +/- the standard error of the mean. A two-tailed *t*-test was used to calculate *p*-values; ***p < 0.01, ***p < 0.001, ****p < 0.0001. NS: not significant. In (c), only the drugs that achieved $p < 0.001$ are indicated.

ELISA kit that detects S100A8 and -A9 only when they are in the functional heterodimer, we found that four agents markedly decreased S100A8/A9 secretion (> 40%) after a 72-hour treatment (Fig. 2b, c). This relatively prolonged treatment length was based on our observation that S100A8/A9 exhibited a protein half-life of 45 ~ 60 hours in MDA-MB-468 cells (Supplementary Fig. 2b), which may be considered unusually long for a cytokine.

Based on the top hits from this screen, we elected to pursue PIM kinase inhibition for further study. The primary rationales for pursuing PIM kinase inhibition were: (i) The relevance of PIM to the biology of TNBC has previously been established in both clinical samples and preclinical models;[19,20] (ii) PIM was recently identified in a highly immunosuppressive population of MDSCs[24], which are a dominant source of secreted S100A8/A9; (iii) Among the top 3-4 hits in our screen, PIM inhibitor PIM447 was most consistent in reducing S100A8/A9 secretion, and these results were readily reproduced using another PIM inhibitor GDC-0339 (Fig. 2c, orange bars), which is considered to have improved pharmacokinetic /pharmacodynamic properties compared to PIM447;[31] (iv) *PIM1,2,3* expression was most significantly and positively correlated with *S100A8* and -*A9* in large cohorts of clinical samples when compared to the intended targets of other inhibitors identified as potential hits in our screen (Supplementary Fig. 2c); (v) Both PIM447 and GDC-0339 reduced S100A8 expression in another S100A8/A9 + TNBC line, BT-20, in a dose-dependent manner (Fig. 2d); and (vi) Our screening results were also reproduced using a genetic approach: MDA-MB-468 cells treated with *PIM1,2,3*-targeted sgRNAs-Cas9 (CRISPR) exhibited significantly reduced expression and secretion of S100A8/A9 (Fig. 2e, f), allowing us to conclude that PIM likely controls *S100A8* and -*A9* expression at the level of transcription (Fig. 2g). Further, we found that lentivirally reintroducing *PIM1* alone to the otherwise *PIM*-null MDA-MB-468 cells restored S100A8/A9 expression and secretion (Fig. 2h, i, and j). These data collectively support that PIM is involved in the mechanism(s) that regulate S100A8/A9 expression.

## Multiple PIM-regulated transcription factors (TFs) control S100A8 and -A9 expression

Our data suggest that PIM could control *S100A8* and -*A9* expression at the transcription level (Fig. 2e, g, h, and j). However, no information in the literature to date associates PIM with S100A8 and -A9. Thus, we utilized the Enrichr program[58,59] and analyzed publicly available datasets, built on ChIP-seq analysis (ENCODE 2015, ChEA2016) and functional genetics methods, to discover putative TFs most reproducibly identified as responsible for *S100A8* and -*A9* expression. Our analysis identified four candidate TFs: STAT3, C/EBPβ, MYC, and PPARG (Fig. 3a and Supplementary Data 3). Among these, STAT3 is regularly supported in the literature as essential for S100A8/A9 expression in both cancer and immune cells[14,60,61], and C/EBPβ has also been shown to control S100A8/A9 expression in a cancer cell model[62]. We found that, specifically in clinical BC samples, *CEBPB* (encoding C/EBPβ) expression was correlated most significantly and positively with that of *S100A8* and -*A9* (Fig. 3b). Interestingly, *CEBPB* was

among the genes associated with poor outcomes in our bioinformatics analysis (Supplementary Data 1).

PIM has been shown to control the activities of STAT3 and MYC through phosphorylation[21,63], and the PI3K/AKT pathway kinases can control C/EBPβ[64,65]. Since PIM and PI3K/AKT pathway kinases are known to share downstream targets[66], we hypothesized that PIM could directly or indirectly control *S100A8* and -*A9* expression by regulating the activity of one of these TFs via phosphorylation. To test this idea, we used siRNA to knock down these TFs in the S100A8/A9-high cell lines MDA-MB-468 and BT-20, which express different levels of the candidate TFs and PIM family members (Fig. 3c). We found that, in MDA-MB-468 cells, knocking down *CEBPB* most potently decreased S100A8 and -A9 protein expression (Fig. 3d, left), whereas knocking down *STAT3* had the most pronounced effect on S100A8 and -A9 abundance in BT-20 cells (Fig. 3d, right).

In a follow-up experiment, we treated these cell lines with PIM inhibitor GDC-0339 for 6 hours and found that GDC-0339 significantly reduced the level of T235-phosphorylated C/EBPβ in MDA-MB-468 cells (Fig. 3e). Interestingly, GDC-0339 also considerably reduced the level of total C/EBPβ (Fig. 3e). These results were reproduced with PIM447 (Supplementary Fig. 3a). Neither the phosphorylation nor the total levels of STAT3 were altered by GDC-0339 within the same time frame in this cell line (Supplementary Fig. 3b). Examining earlier time points (~60 min) showed that PIM inhibition-induced T235-dephosphorylation preceded a reduction in total C/EBPβ levels (Supplementary Fig. 3c). Based on these observations and recent findings that the ubiquitin-proteasome system (UPS) controlled protein abundance of C/EBPβ[67,68], we hypothesized that PIM inhibition-induced T235-non-phosphorylated C/EBPβ is degraded by the UPS. To test this hypothesis, we treated MDA-MB-468 cells with GDC-0339 in the presence of a low-dose general proteasome inhibitor (MG132) and found that PIM inhibition did not reduce total C/EBPβ levels when the proteasome was inhibited (Fig. 3e). Next, we immunoprecipitated total C/EBPβ from the cells treated with GDC-0339 and performed anti-ubiquitin western, which showed that PIM inhibition rapidly and markedly increased the abundance of polyubiquitinated C/EBPβ (Fig. 3f). These observations are reconcilable with our earlier screening data showing that carfilzomib, a clinical proteasome inhibitor, notably increased S100A8/A9 secretion in MDA-MB-468 cells (Fig. 2c). In BT-20 cells, GDC-0339 treatment resulted in dephosphorylation of STAT3 S727 (Fig. 3g), which was reproduced with PIM447 (Supplementary Fig. 3d), and PIM inhibition did not affect the abundance or phosphorylation status of C/EBPβ in the time frame that the drug treatment was conducted (Supplementary Fig. 3e). Interestingly, our attempt to reproduce these cellular phenotypes using a genetic method revealed that both C/EBPβ and STAT3 were markedly altered in the aforementioned MDA-MB-468 CRISPR-*PIM1,2,3* cells, and these phenotypes could be mostly reversed by exogenously reintroducing *PIM1* alone (Fig. 3h). Thus, although most of our data favor the idea that a single PIM-regulated TF (i.e. C/EBPβ or STAT3) serves as the primary TF for *S100A8* and -*A9* in a given context (Fig. 3i), it remains unaddressed

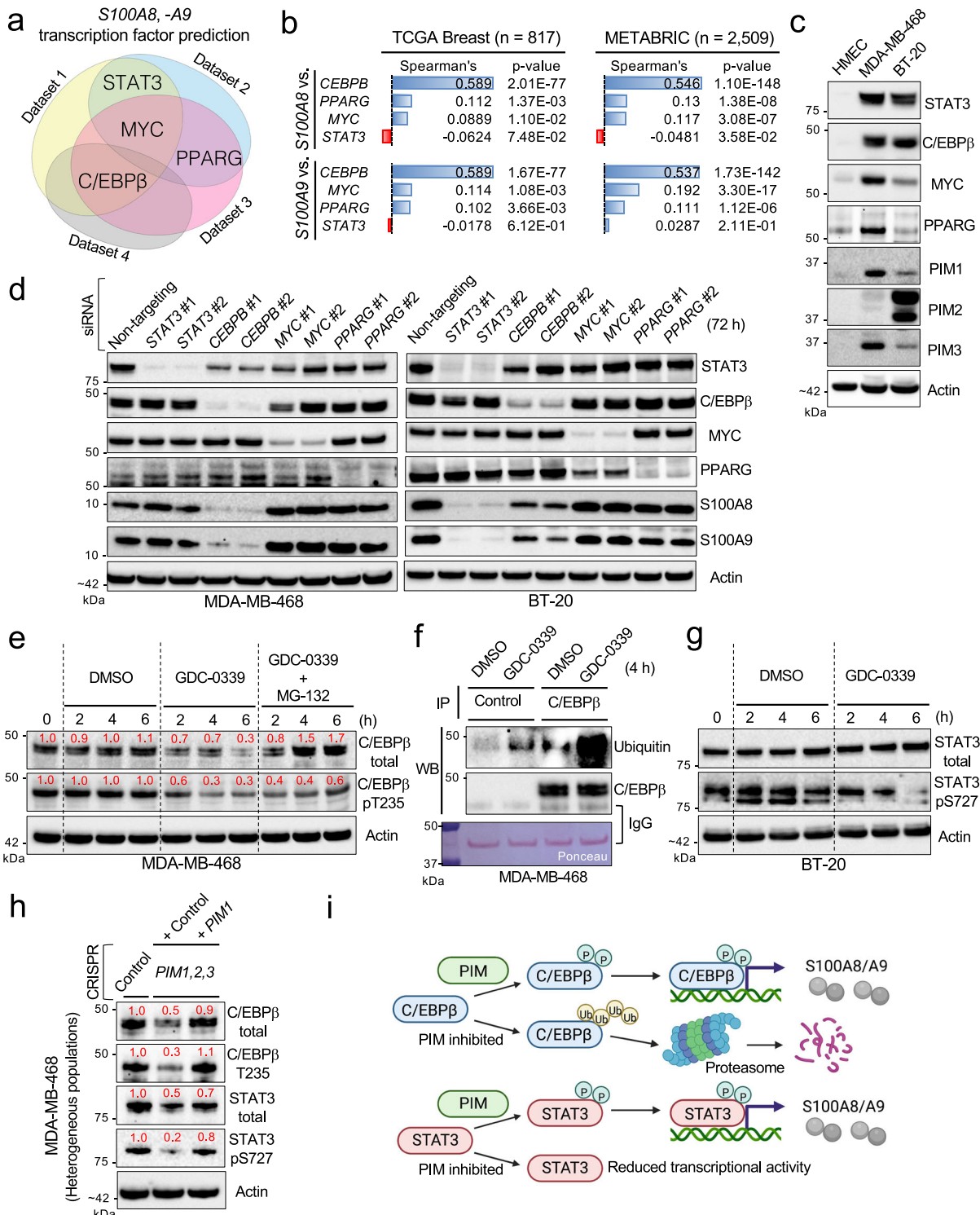

**Fig. 3 | PIM controls S100A8 and -A9 expression by regulating C/EBPβ and STAT3. a** Venn diagram showing the transcription factors (TFs) most reproducibly identified as responsible for *S100A8* and *-A9* expression across four public datasets (ENCODE 2015, ChEA 2016, TF Perturbations/Expression, and TRUST TF 2019) in the Enrichr database (Supplementary Data 3). **b** Gene co-expression analysis of *S100A8* or *-A9* and their candidate TFs in the TCGA breast and breast cancer METABRIC datasets. **c** Representative western blots showing protein expression levels of the candidate TFs identified in (**a**) and PIM1, -2, and -3 in the indicated TNBC cell lines and HMEC cells. **d** Representative western blots showing the effects of knocking down the candidate TFs identified in (**a**) on S100A8 and -A9 expression in the indicated TNBC cell lines. Two siRNA sequences per gene were used as indicated. **e** Representative western blots showing the effects of GDC-0339 alone or in the presence of MG-132 (1 μM) on the levels of total or T235-phosphorylated C/EBPβ. **f** Representative western blots showing the effects of GDC-0339 on the abundance of ubiquitinated C/EBPβ. IP: immunoprecipitation; WB: western blotting. **g** Representative western blots showing the effects of GDC-0339 on the levels of total or S727-phosphorylated STAT3. **h** Representative western blots showing the levels of total and phosphorylated C/EBPβ and STAT3 in the cells transiently transfected with Cas9/*PIM1,2,3*-specific sgRNAs, in the presence or absence of exogenously introduced PIM1. **i** Schematic representation of the proposed signaling mechanisms by which PIM controls S100A8 and -A9 expression. Throughout this figure, GDC-0339 was used at 5 μM. Actin serves as a loading control. The numbers in red indicate relative protein expression.

whether multiple TFs may at some point become interdependent in maintaining pro-tumorigenic levels of S100A8 and -A9.

## PIM inhibition sensitizes otherwise resistant S100A8/A9-high tumors to ICIs

Emerging clinical data showing that increased S100A8/A9 expression correlates with resistance to anti-PD-1 therapy in solid cancer[17] has begun substantiating a slew of preclinical evidence demonstrating how S100A8/A9 may contribute to generating immunosuppressive TME. Thus, based on our observation that PIM is required for the expression of S100A8/A9, we hypothesize that PIM inhibition could aid in sensitizing particularly S100A8/A9-high tumors to ICIs. To test this hypothesis, we set out to determine whether PIM inhibition, when combined with immune checkpoint blockade, induces superior antitumor responses in an S100A8/A9-dependent manner (Fig. 4a). To this end, we utilize three syngeneic TNBC cell lines to model various expression levels of S100A8, -A9, PIM (Fig. 4b), PD-1, PD-L1 (Fig. 4c), T cell infiltration (Supplementary Fig. 4a), and MDSCs (Supplementary Fig. 4b). MDSCs were identified using a recently revised set of markers, including CD84[11,24] (Supplementary Fig. 4c), to distinguish them from monocytes and neutrophils[24].

First, we examined the ability of PIM inhibitors to modulate their downstream targets in mouse myeloid cells in vitro in the same manner that they did in human TNBC cells (Fig. 2d, Fig. 3e, g, and Supplementary Fig. 3a, c, and d). The rationale for this is that non-cancer cells, particularly myeloid cells, are the dominant sources of S100A8/A9 in developed tumors in mouse models of BC (Ref. [16] and Supplementary Fig. 4d). The D2A1-M1 line[49] (S100A8/A9-high, Fig. 4b) was orthotopically injected into the fourth mammary glands of female mice for tumor formation. CD11b + Gr1+ myeloid cells were then isolated from larger tumors (approximately 1000 mm³, which enabled the acquisition of enough cells for subsequent in vitro assays) and treated with GDC-0339 and PIM447. We found that both inhibitors markedly altered the protein abundance of S100A8 and -A9, as well as total and phosphorylated C/EBPβ and STAT3 (Fig. 4d).

Next, we investigated the potential of a combination therapy targeting both PIM and PD-1 or PD-L1 to either inhibit continued tumor growth, or to induce tumor regression in a manner proportional to tumoral S100A8/A9 levels. D2A1-M1, -M2[49], and Py8119[69] (S100A8/A9-high, -medium, and -low, respectively, Fig. 4b) were utilized in this efficacy study. For D2A1-M1 and -M2 lines, tumor-bearing mice were treated for two weeks with GDC-0339 alone (100 mg/kg, daily, six days/week: the maximum tolerated dose is approximately 300 mg/kg[31]), anti-PD-1 or -PD-L1 antibody alone (12 mg/kg, twice a week/3 days apart), or GDC-0339 and anti-PD-1 or -PD-L1 antibody combined (Fig. 4a). We found that the combination therapy substantially inhibited the continued growth of fast-growing D2A1-M1 tumors (Fig. 4e, left) while inducing tumor regression in some mice (Fig. 4e, middle). D2A1-M2 tumors, compared to D2A1-M1 tumors, were less sensitive to combination therapy. However, this combination still elicited significant growth inhibition in these tumors (Fig. 4f). Combining GDC-0339 and an anti-PD-1 antibody did not significantly affect the continued growth of Py8119 tumors (Fig. 4g). Thus, as observed in these three syngeneic orthotopic models, the differences in sensitivity are likely best explained by the combination of overall S100A8/A9 abundance (Fig. 4b), the gross abundance of MDSCs (Supplementary Fig. 4b), and PD-1/PD-L1-positivity among CD45- cells (Fig. 4c) in tumors. One limitation of our analysis is the exclusion of other potentially relevant myeloid cell types, such as M2 macrophages and tumor-associated N2 neutrophils, due to the challenges in reproducibly identifying them by flow cytometry alone[70–73].

We subsequently subjected the drug-treated D2A1-M1 tumors to the NanoString Immuno-Oncology 360 (IO360™) platform (NanoString Technologies, Inc.) which showed that, among the 770 IO-related genes in this panel, *S100A8* was one of the most markedly down-regulated genes in tumors treated with GDC-0339 alone (Supplementary Fig. 4e). Although the FDR-adjusted *p*-value did not indicate statistical significance in this cohort, our follow-up qPCR validation experiment showed that GDC-0339 significantly reduced *S100A8* expression, and considerably reduced *S100A9*,

albeit not in a statistically significant manner (Supplementary Fig. 4f). Through its proprietary scoring method[74], the IO360™ platform also revealed that GDC-0339, in combination with an anti-PD-1 or -PD-L1 antibody, moderately increased the abundance of tumor-infiltrating total and non-exhausted CD8 + T cells, resulting in a significantly higher "cytotoxic cells score" (Fig. 4h). This score is indicative of the abundance of cytotoxic molecules such as granzyme B, perforin, and TNF-alpha (Fig. 4i)[74]. These data collectively support the idea that, when combined with immune checkpoint blockade, PIM inhibition can incite cytotoxic immune responses, especially in S100A8/A9-high tumors. To further confirm the significance of increased cytotoxic immune responses in D2A1-M1 tumors upon combination treatment with GDC-0339 and anti-PD-1 antibody, we simultaneously treated with an anti-CD8 neutralizing antibody to deplete CD8 + T cells. We found that the inhibition of continued growth of D2A1-M1 tumors was significantly compromised in the presence of an anti-CD8 antibody (Supplementary Fig. 4g).

## PIM inhibition-induced early changes include a decreased abundance of tumoral MDSCs

Our in vivo efficacy experiments suggest that PIM inhibition can sensitize otherwise resistant S100A8/A9-high TNBC tumors to immune checkpoint blockade (Fig. 4e, f, h, and i). However, the NanoString data are confounded by vast differences among tumor sizes at the end of the 2-week treatment period (i.e., Fig. 4e) and do not elucidate the stage of combination treatment that PIM inhibition might be most critical for tumor growth inhibition or regression to occur. To address this issue, we conducted a short-term, 5-day GDC-0339 treatment in mice (Fig. 5a) and asked the following questions.

First, we asked whether the effect of PIM inhibition on *S100A8* and -*A9* expression observed in vitro (Fig. 2g) and at the end of the 2-week treatment in vivo (Supplementary Fig. 4e, f) could be reproduced at an earlier time point in vivo. Mice with D2A1-M1 tumors (75-100 mm³) were treated with GDC-0339 at 100 mg/kg daily. Significant reductions in *S100A8* and -*A9* mRNA expression were observed in the tumors isolated from mice that received only five doses of GDC-0339 (Fig. 5b). During the 5-day GDC-0339 treatment, we also asked whether S100A8/A9 levels in serum could be i) determined reliably and reproducibly, ii) whether they corresponded to S100A8/A9 levels in tumors, and iii) whether serum S100A8/A9 could thus serve as a pharmacodynamic biomarker to monitor tumors' responses to GDC-0339. We found that serum S100A8/A9 was readily detectable in tumor-bearing mice and that, as tumors grew from 75-100 mm³ to approximately 200 mm³ in the 5-day treatment period (e.g., Fig. 4e), serum S100A8/A9 levels rose significantly in the vehicle group, whereas no such increase was seen in the serum of their GDC-0339-treated counterparts (Fig. 5c). As such, our data show that serum concentrations do in fact reflect the GDC-0339-altered expression of *S100A8* and -*A9* in D2A1-M1 tumors (Fig. 5b). Next, we asked whether this GDC-0339-induced reduction in day 5 tumor *S100A8* and -*A9* and serum S100A8/A9 levels is associated with changes in the abundance of select immune cell types in tumors, namely CD8 T cells, Tregs, and MDSCs. We found that, while the 5-day GDC-0339 treatment did not result in statistically significant changes in the abundance of CD8+ and Treg cells (Fig. 5d), it did significantly reduce the abundance of total MDSCs (Fig. 5e). Finally, we investigated the potential correlation between S100A8/A9 levels in serum samples from TNBC patients and different stages of the disease. We found that the samples from Stage 1 patients exhibited varying levels of S100A8/A9, as did those from Stage 3 and 4 patients, with some samples exceeding 5.5 µg/mL (Fig. 5f), a level associated with poor responses to anti-PD-1 therapy in a recent melanoma study[17]. There were no detectable levels of S100A8/A9 in the samples from healthy donors. These results suggest that serum S100A8/A9 levels may serve as a predictor of tumor responses to specific treatments and subsequent disease progression (i.e., Fig. 1a), rather than reflecting breast cancer stages.

To further model how the presence or absence of secreted S100A8/A9 affects the TME, we conducted an in vitro chemotaxis assay using human PBMC-derived immune cells (Fig. 5g). We found that conditioned media

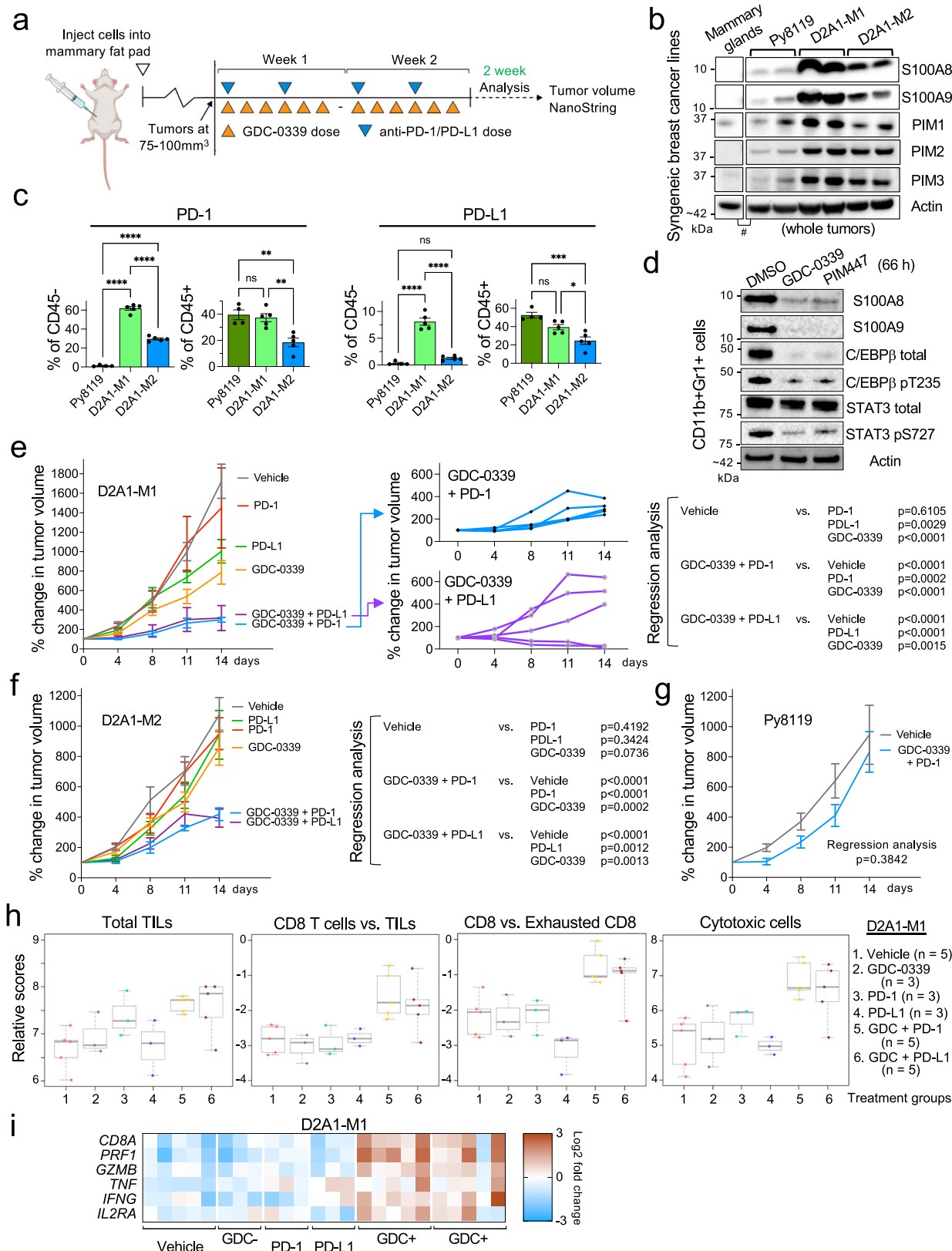

from MDA-MB-468 CRISPR-*S100A8* cells, which lacked S100A8/A9 (Supplementary Fig. 5a), exhibited a significantly reduced ability to recruit immunosuppressive myeloid cells (Fig. 5h, control vs. *S100A8*-KO) without altering the ratio of various migrating cell types, such as MDSCs and neutrophils (Fig. 5i). These results are similar to those previously obtained using mouse tumor-derived materials and S100A8 and -A9-blocking antibodies[15]. In the same assay, we also found that conditioned media from MDA-MB-468 CRISPR-*PIM1,2,3* cells nearly lost the ability to induce cell migration and that such an effect could be reversed by exogenous *PIM1* expression (Fig. 5h). Interestingly, the same conditioned media did not affect the

**Fig. 4 | Small-molecule PIM inhibition as a tool to sensitize otherwise resistant S100A8/A9-high PD-1/PD-L1-positive tumors to immune checkpoint inhibitors. a** Schematic representation of the flow of the experiments and downstream analyses conducted. **b** Representative western blots showing protein abundance of S100A8, -A9, PIM1, -2, and -3 in the indicated tumors (two samples per line) or non-tumor tissues. #This panel is a composite of two sides of identical western blot. **c** Flow analysis showing PD-1/PD-L1-positivity in the indicated cell populations in size-controlled tumors (500 mm³; $n = 4$ for Py8119, $n = 5$ for D2A1-M1 and -M2). **d** Representative western blots showing the effects of the indicated pan-PIM inhibitors (5 μM) on the total abundance or phosphorylation levels of the indicated proteins in CD11b + Gr1+ myeloid cells isolated from D2A1-M1 tumor-bearing mice. **e, f** Growth of indicated tumors in mice treated with the indicated drugs or drug combinations for 2 weeks ($n = 5$ per treatment group). Linear regression analysis was used to calculate $p$-values. **g** Growth of Py8119 tumors in mice treated

with GDC-0339 plus an anti-PD-1 antibody ($n = 5$ per treatment group). **h** The effects of the indicated drugs or drug combinations on the indicated cytotoxic immune response-related parameters (scoring system) adopted by the NanoString IO360™ platform. Box plots show the 25th, 50th (median), and 75th percentiles. These parameters have not been designed to generate data indicative of statistical significance. TILs, tumor-infiltrating lymphocytes. **i** Heat map representation of the effects of the indicated drug treatment on the relative expression of the indicated cytotoxic immune response-related genes. The individual tumor samples represented in this heat map correspond to those in Fig. 4h. Throughout this figure, GDC-0339 was used at 100 mg/kg/dose, and anti-PD-1/PD-L1 antibodies were used at 12 mg/kg/dose in vivo. Error bars represent means +/- the standard error of the mean. Unless otherwise indicated, a two-tailed $t$-test was used to calculate $p$-values; *$p < 0.05$, **$p < 0.01$, ***$p < 0.001$, ****$p < 0.0001$. NS: not significant. Actin serves as a loading control.

proliferation and activity levels of CD8 + T cells, Tregs, and MDSCs differentially (Supplementary Fig. 5b, c). One exception was that CD8 + T cells exposed to the conditioned media from MDA-MB-468 CRISPR-*PIM1,2,3* cells produced more TNF-α in an inconspicuous but statistically significant manner (Supplementary Fig. 5c, TNF-α+ of CD8). These data suggest that PIM may play a pivotal role in regulating the expression or secretion of various yet-to-be-identified cytokines and chemokines, in addition to S100A8/A9, that are critical for facilitating robust tumor infiltration of immunosuppressive myeloid cells. Thus, it may be rational to infer that early-stage PIM inhibitor-induced changes in the abundance of immunosuppressive myeloid cells (i.e. MDSCs, Fig. 5e)—which could reasonably be expected to influence the activities of other immunosuppressive cell types such as Tregs[75,76]—represent some of the prerequisite events for potentiating subsequent cytotoxic immune responses.

## Discussion

Compared with other BC types, TNBC is strongly associated with higher levels of tumor-infiltrating lymphocytes (TILs)[77], making TNBC a promising candidate for treatment with established ICIs. Indeed, pembrolizumab (anti-PD-1 antibody) has recently been FDA-approved for TNBC[78,79], and atezolizumab (anti-PD-L1 antibody), though more recently withdrawn from further regulatory considerations in the US, initially generated favorable clinical responses[80]. Despite this enthusiasm, the clinical response for currently adopted forms of immunotherapies has not been satisfactory (e.g., the median overall survival among patients with advanced TNBC highly positive for PD-L1 was 23.0 months when treated with pembrolizumab plus chemotherapy, and 16.1 months for placebo plus chemotherapy)[78,81]. Thus, there is considerable interest in identifying resistance mechanisms for and superior predictive biomarkers of response to ICIs in TNBC. The present study aimed to comprehend the biological basis of poor clinical outcomes in TNBC and identified S100A8/A9 as one of the factors whose increased expression in early-stage tumors was most strongly associated with the risk of subsequent disease progression. The gene expression datasets used in this study, accompanied by reliably extended clinical follow-up information (Supplementary Fig. 1b, c), were primarily based on neoadjuvant chemotherapy and not on any immunotherapies. Despite this, it seems reasonable to postulate that one of the reasons patients with S100A8/A9-high TNBC tumors progressed on neoadjuvant chemotherapy may be a dysregulated immune response (Fig. 1c). Indeed, emerging clinical evidence indicates that abundance and activity levels of TILs are indicative of tumors' response not only to immunotherapies but also to cytotoxic chemotherapy in TNBC patients[82]. Furthermore, the abundance of immunosuppressive myeloid cells has recently been identified as significantly correlated with resistance to ICIs[83,84]. In this respect, factors that can be established as substantial contributors to generating or maintaining an immunosuppressive TME, regardless of what prognostic parameters are used to identify them, will likely be promising candidates to target together with the PD-1/PD-L1 axis in developing effective combination therapies.

Our chemical genetic approach employing targeted small molecule screening and genetic validation identified PIM kinase inhibition as a clinically viable method to decrease S100A8/A9 expression and secretion (Fig. 2). Thus, the data presented in this report, combined with a wealth of existing preclinical functional data and emerging clinical observations on S100A8/A9, seem to support the proposition that the combination therapy targeting PIM and the PD-1/PD-L1 axis can be investigated clinically in TNBC using S100A8/A9 as a predictive and pharmacodynamic biomarker. Although countless clinical trials have evaluated the efficacy of ICIs in combination with numerous targeted anticancer agents[85], a mechanistic understanding of how these combinations could reasonably be expected to induce drug synergies in the TME is not yet understood. Furthermore, often lacking from these trials are mechanism-based biomarker and patient stratification strategies. In this context, considering that S100A8/A9 as a serum biomarker seems to withstand the rigors of clinical investigations[17], our proposed approach of incorporating S100A8/A9 not only as a therapeutic target but also as a biomarker for developing therapeutic strategies in TNBC may prove advantageous both scientifically and operationally. For its utility as a serum biomarker to be established in TNBC, however, whether serum S100A8/A9 levels reproducibly correlate with those of tumoral S100A8/A9 in patients' samples will need to be determined longitudinally (i.e., Fig. 5b, c).

In summary, the present study reports identifying increased S100A8/A9 expression in early-stage tumors as one of the factors most strongly associated with subsequent disease progression in TNBC. Additionally, this study reveals a previously unappreciated utility of PIM kinase inhibition in reducing S100A8/A9 expression and secretion and tumor infiltration of immunosuppressive myeloid cells in an S100A8/A9-dependent and -independent manner. To the best of our knowledge, it shows for the first time using orthotopic solid cancer mouse models that PIM inhibition, when combined with immune checkpoint blockade, can induce significant antitumor effects, especially in S100A8/A9-high PD-1/PD-L1-positive tumors that may be otherwise resistant to either therapy alone (Fig. 6). Although this study and a recent report[25] share a general idea that PIM inhibition can be used to counteract immunosuppressive TME, these two studies diverge entirely in the proposed mechanisms underlying this effect, the previous report focusing on controlling fatty acid metabolism in myeloid cells, governing their suppressive capabilities. Finally, it was previously shown that most xenografted mouse tumors tested continued to grow, albeit at significantly slower rates, in the absence of S100A8/A9[12,14,16]. Therefore, it is critical to acknowledge that, when combined with immune checkpoint blockade, the capacity of PIM inhibition to induce significant antitumor responses in S100A8/A9-high tumors (Fig. 4e, f, h, and i) is not solely due to its ability to decrease S100A8/A9 expression in the TME. Rather, it is most decidedly due to its ability to simultaneously act on well-characterized cancer cell-intrinsic growth-promoting pathways, such as the MYC pathway active in TNBC[86], the mechanisms that control S100A8/A9 expression in multiple cell types in the TME, and yet-to-be-explored mechanisms by which PIM may control expression or secretion of additional cytokines and

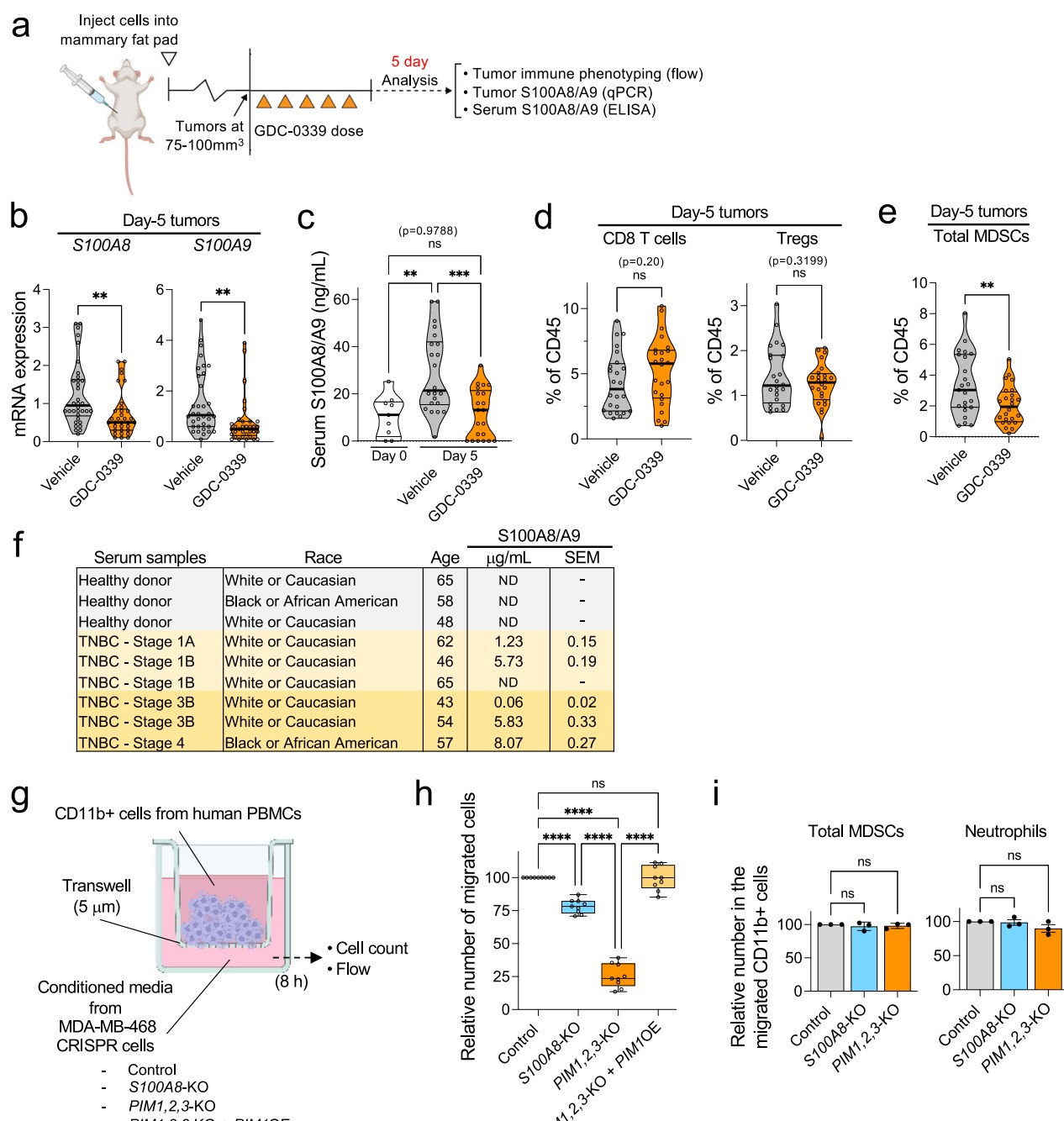

**Fig. 5 | The effects of short-term GDC-0339 treatment on the abundance of S100A8/A9 and select immune cell types in vivo. a** Schematic representation of the flow of the experiments and downstream analyses conducted. **b** Relative mRNA levels of *S100A8* and *-A9*, as determined by qPCR, in the D2A1-M1 tumors treated with GDC-0339 or vehicle for five days ($n = 34$ for the vehicle group, $n = 33$ for the GDC-0339 group). **c** Changes in serum S100A8/A9 levels, as determined by ELISA, in D2A1-M1 tumor-bearing mice treated with GDC-0339 or vehicle for 5 days ($n = 9$ for Day 0 samples, $n = 22$ for the vehicle group and $n = 21$ for the GDC-0339 group for Day 5 samples). **d** Flow analysis showing the abundance of the indicated T cell types in tumors isolated from D2A1-M1 tumor-bearing mice treated with GDC-0339 or vehicle for 5 days ($n = 24$ per treatment group). **e** Flow analysis showing the total abundance of myeloid-derived suppressor cells (MDSCs, polymorphonuclear- and monocytic-MDSCs combined) in tumors isolated from D2A1-M1 tumor-bearing mice treated with GDC-0339 or vehicle for 5 days ($n = 23$ for the vehicle group, $n = 24$ for the GDC-0339 group). **f** Serum S100A8/A9 concentrations, as

determined by ELISA, in samples from healthy donors and patients with TNBC at the indicated breast cancer stages. Each sample was subjected to the assay three times independently, and the concentrations shown represent the average values with the standard error of the mean (SEM). All donors were females. ND: not detectable (concentrations <approximately 0.002 µg/mL). **g** Schematic representation of the in vitro chemotaxis assay performed. PBMCs: peripheral blood mononuclear cells. KO: knockout. OE: overexpression. **h** Box plots showing the effects of conditioned media from the indicated MDA-MB-468 CRISPR lines on CD11b+ cell migration (8 h) ($n = 9$ per experimental group). **i** Relative abundance of total MDSCs (left) or neutrophils (right) in the migrated populations of CD11b+ cells as shown in Fig. 5h ($n = 3$ per experimental group). Throughout this figure, GDC-0339 was used at 100 mg/kg/dose. Error bars represent means +/- SEM. Thick black lines indicate the median in violin plots. A two-tailed *t*-test was used to calculate *p*-values; ***p* < 0.01, ****p* < 0.001, *****p* < 0.0001. NS: not significant.

**Fig. 6 | Proposed mechanisms of drug synergy between PIM kinase inhibition and immune checkpoint blockade in TNBC.** Schematic representation of how a small-molecule pan-PIM kinase inhibitor, when combined with immune checkpoint inhibitors (ICIs) such as anti-PD-1 and PD-L1 antibodies, can be expected to contribute to the regression of S100A8/A9-high PD-1/PD-L1-positive TNBC tumors.

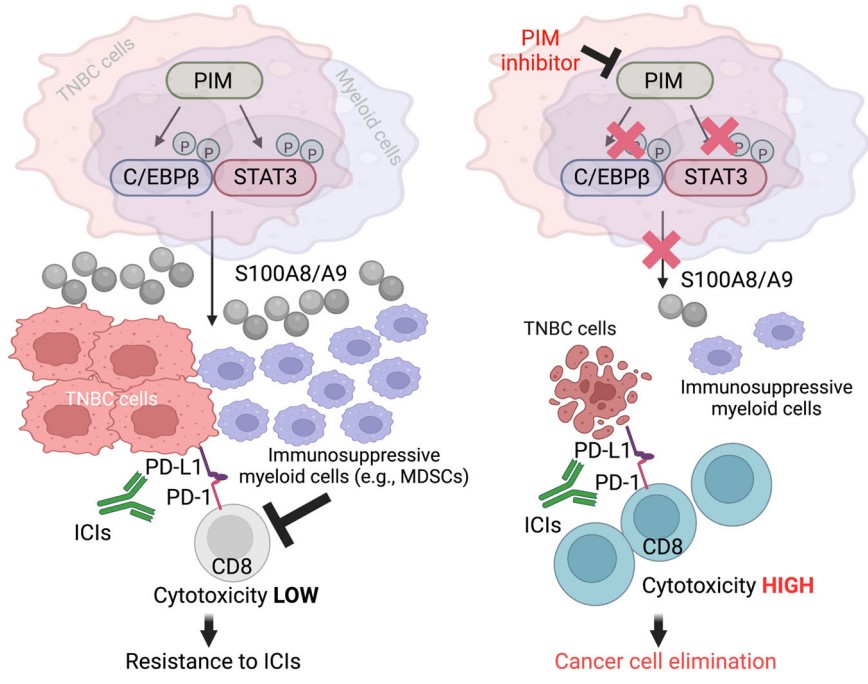

chemokines critical for recruiting various immunosuppressive myeloid cell types (Figs. 5g, h, and 6).

## Data availability

All necessary data to assess the conclusions in this manuscript are included in the manuscript and supplementary files. Supplementary Figs. 6-14 contain the original digital images of chemiluminescence signals used in protein expression analysis. Source data for Figs. 1–5 and Supplementary Figs 3-5 are available in separate supplementary data files accompanying this manuscript (Supplementary Data 1-5). All other data are available from the corresponding author on reasonable request.

## Code availability

The custom computer code used to identify the prognostic gene set comprising 210 genes (Supplementary Data 1) can be found on GitHub (https://github.com/DHBCIOLAB/S100A8A9_TNBC.git).

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

## Acknowledgements

We acknowledge the following support: the Lynn Sage Breast Cancer Foundation (A.V., D.H.), the Elsa U. Pardee Foundation (A.V., D.H.), the Northwestern Medicine Catalyst Funds (D.H.), the US National Institutes of Health (NIH; R01CA258833 to D.H.), the US Department of Defense Breast Cancer Research Program (W81XWH1810053 to D.H.), the Susan G. Komen Foundation (CCR16376693 to D.H.), the Translational Bridge Program of Robert H. Lurie Comprehensive Cancer Center of Northwestern University (D.H.), the Northwestern University Clinical and Translational Sciences Institute (D.H.), which was supported by the NIH's National Center for Advancing Translational Sciences (UL1TR001422). Flow cytometry experiments were carried out at the Northwestern University Flow Cytometry Core Facility, and a part of the small molecule screening was carried out at the Northwestern High Throughput Analysis Laboratory, both of which were partly supported by a National Cancer Institute Cancer Center Support Grant (P30CA060553). The content is solely the authors' responsibility and does not necessarily represent the official views of the NIH. Some of the graphical figure panels were created with BioRender.com. We thank Drs. Clare Isacke and Ute Jungwirth for providing the D2A1-M1 and -M2 lines, Levi Barse, Dr. Erica Fleming-Trujillo, and Miriam Walter for the initial characterization of the experimental models used in this study, Dr. Kai Kessenbrock for his insights into flow cytometry-based identification of MDSCs and the potential significance of targeting PIM kinases in MDSCs, Genentech for providing GDC-0339, Dr. Andrei Goga for supporting S.B. and R.C. during the construction of the initial version of bioinformatics pipeline and D.H's transition to independence, and Dr. J. Michael Bishop and Laura Sage for their encouragement.

## Author contributions

L.R.B., A.M.O., M.Z., P.V., S.F.D., A.V., T.H.H. and D.H. contributed to designing, executing, and interpreting data from in vitro experiments. L.R.B., A.M.O., D.K., R.B., N.A.Z., A.V., T.H.H. and D.H. contributed to designing, executing, and interpreting data from in vivo experiments. C.Yeh, S.B., R.C., C.Yau., K.A.K. and D.H. contributed to bioinformatics analyses. D.R. served as the Horiuchi lab's primary breast cancer patient advocate, provided guidance from the patient's perspective, and helped the team secure research funding. L.R.B., A.M.O., S.B., R.C., C.Yau, T.H.H., K.A.K. and D.H. participated in writing the paper. D.H. conceived the research question, provided funding for, and supervised the study.

## Competing interests

During this study, T.H.H. was affiliated with Cancer Immunology Discovery, Pfizer, Inc., San Diego, CA, as a full-time employee and participated in this research in his capacity as an adjunct faculty member of Northwestern University's Feinberg School of Medicine. The authors declare no competing non-financial interests, but the following competing financial interests: During this study, S.F.D. held equity interests (> US $5,000) in AbbVie and Abbott, T.H.H. held equity interests in Pfizer, and D.H. held equity interests in Merck, Eli Lilly, and AstraZeneca.
