## [Peer Review File · Communications Medicine]

Reviewers' comments:

Reviewer #1 (Remarks to the Author):

The manuscript is well-written and provides a substantial amount of data. There are instances where the writing in the introduction can be improved. There are mixture of ideas presented within one paragraph.

Line 68-73 - the statement describing the overall goal of the manuscript is not a typical component of the introduction and should be removed. The same is true for line 91-94.

Reviewer #2 (Remarks to the Author):

The aim of this work was identifying clinically viable therapeutic strategies in TNBC and potential biomarkers of response. In particular, in this manuscript the authors describe how PIM kinases are involved in the regulation of S100A8/A9 expression in TNBC and demonstrate the potential of targeting PIM in combination with anti-PD-1 antibodies to reduce tumor growth and increase cytotoxic immune responses in preclinical models. Overall, the rationale supporting the studies is well defined, the choices in the experimental design are properly justified and the statistical analysis is complete. Thus, this study provides relevant and mechanistic knowledge on how S100A8/A9 regulates the immune component of the TME in vitro and preclinical models. Therefore, only a few aspect require further revision:

Major comments:

1. Line 119: The prognostic potential of S100A8/A9 is evident with the background provided by the authors. However, authors should clarify and elaborate on the use of S100A8/A9 as a predictive biomarker and specify the treatment for which it could be considered a predictive biomarker.
2. To support the choice of the experimental cell model in figure 2 (MDAMB468 cells), authors should consider including available data of the aggressiveness and response to treatments of the 4 TNBC cell lines analyzed (e.g. in preclinical models) and discuss the differences between the cell lines expressing S100A8/A9 and the ones that do not express S100A8/A9.
3. Given the scope of Communications medicine and to support the potential use of S100A as a biomarker (that the authors highlight in the abstract, in the last paragraph of the introduction as well as in the results section – Fig 4), some evidence should be provided in the context of breast cancer patient samples. In particular, as discussed by the authors, S100A8/A9 could be used as a serum biomarker. As the authors indicate, the correlation of serum S100A8/A9 levels with those of tumor samples should be determine in future studies. However, adding evidence of S100A8/A9 detection in blood samples of breast cancer patients and its association with outcome will already strengthen the translational potential of their findings.

Minor comments:

1. Line 56: Clarify that those findings were obtained in mouse models.
2. Line 77 and 167: Add reference.
3. Line 155: To provide a complete analysis, authors could add the Kaplan-Meier graphs for the other 2 cohorts not included in Figure S1B
4. Line 161: HER2-positive tumors exhibit elevated S100A8 and A9 expression in all the cohorts, not in some.
5. Line 259: add "expression" before "levels"

Reviewer #3 (Remarks to the Author):

Summary of findings:

In this manuscript, Begg et al. set out to unravel why some TNBC tumors progress despite existing therapies. To this end, they performed several bio-informatic meta-analyses on four publicly available datasets using semi-supervised approaches. The study identified 30 genes that were dysregulated in at least 3 of the 4 datasets, among which S100A9 came out as the most prognostic of the upregulated genes. A subsequent, less stringent analysis also identified the binding partner, S100A8 as a prognostic marker. The authors next performed a drug screening in a S100A8/A9-expressing TNBC cell line to identify compounds that modulate the expression of S100A8/A9. Most notably, PIM-inhibitors (PIM447, GDC-0339) consistently lowered the levels of heterodimeric S100A8/A9. This hit was subsequently validated using a series of convincing ELISA, Western blot and genetic CRISPR approaches, supporting that PIM is involved in the expression of S100A8/A9. Further mechanistic insight was gained by computationally inferring putative transcription factors that modulate S100A8/A9 expression (STAT3 and C/EBP β) and using a combination of knock-out experiments to evaluate how these are modulated by PIM. Finally, the authors use PDX mice models to demonstrate that PIM inhibition sensitizes S100A8/A9-high tumors to anti-PD1/PD-L1 antibody IC blockade, showing increased infiltration of cytotoxic T cells. Interestingly, the authors provide some evidence that S100A8/A9 levels in serum mirror the abundance in tumors, suggesting a potential application as a liquid biopsy biomarker.

Overall assessment of quality and importance:

This is a well-written, hypothesis-driven manuscript with rigorous methodology. There is indeed a lack of understanding how some TNBC tumors progress despite existing therapies, while others do not. This manuscript builds on previous knowledge that S100A8/A9 can induce an immunosuppressive TME in other cancer types. Importantly, this study also aimed to unravel the mechanisms behind this process and nominates PIM as an important player in S100A8/A9 expression. As such, this manuscript provides fundamental knowledge that is potentially applicable beyond the setting of TNBC. Although the applicability of S100A8/A9 in the serum as a biomarker needs further validation in a clinical context, as mentioned by the authors, this is an interesting aspect to the study indeed. Overall, I commend the authors for their work and support publication in Nature Communications Medicine. I only have a few minor comments, listed below.

Minor comments:

- The authors describe that S100A8/A9 can be expressed by several cell types, including cancer cells, cells of the myeloid lineage and TAMs. It is also indicated that in the murine models of TNBC, the major source of S100A8/A9 is cells of myeloid origin. As the study mainly focuses on the expression of S100A8/A9 in cancer cells, it would be informative to comment on the relative contribution of cancer cells to S100A8/A9 levels in the TME, compared to other cell types. Are these the major source of S100A8/A9 in the TME in a human context?
- I support the idea of a consensus approach to identify prognostic markers. However, I would have liked to see a summary table listing the main characteristics of the four datasets used in the study, in order to evaluate the variance among them. Especially parameters like sample size, gender distribution, age ranges would be useful to compare the different studies at a glance.
- Did you start from processed count tables or raw fastq files of the four datasets? If starting from

count tables, some comment on the influence of the pre-processing steps (QC, mapping, etc.), which probably differed among the studies, on the final data would be in place.

- Regarding the group of 30 genes that were most robustly associated with disease progression: are the genes with the highest significance scores (GATA3, NAT1, ...) known to be involved in TNBC disease progression? If so, this would validate the approach, giving more confidence to the S100A9 hit. Please elaborate.
- Regarding the gene-wise bootstrap scores used for sample stratification (lines 421-423), it would be useful to provide the readers with the ranked list for future reference.
- On the same topic, I would be interested to see which genes are in the optimal multi-gene classifier (lines 431-432), which was used to stratify the samples. How many genes were used in the end? Which genes? Are S100A8/A9 or other hit genes part of the classifier?
- Some TNBC cell lines are high in S100A8/A9, while others do not seem to express this. Is anything known about how these behave differently in PDX models? One would expect the S100A8/A9-low cell lines to progress slower compared to the S100A8/A9-high cell lines. Please comment.

We extend our heartfelt appreciation to the reviewers for their invaluable time and effort in providing exceptionally constructive and thoughtful comments and suggestions.

Before addressing the individual comments of the reviewers, we would like to draw their attention to a modification we have implemented in this revised manuscript. In the previous version of the manuscript, we presented and referred to our genes of interest in two different ways throughout the text, including the supplementary datasets. One group consisted of 30 genes resulting from the analysis of four datasets, while the other encompassed 210 genes derived from the analysis of three datasets. Although the “clustered bubble chart” in the previous version of Fig. 1b illustrated the former (30 genes), the remainder of the manuscript, including critical biological pathway analysis (Fig. 1c), mechanistic studies (Fig. 3, i.e., C/EBP β), in vivo efficacy studies (such as cytokine signaling and the tumor microenvironment), and the discussion section, heavily relied on the 210 genes. To eliminate this inconsistency and prevent confusion for future readers, we have replaced the “clustered bubble chart” with a volcano plot, illustrating the potential relative significance of the identified 210 genes. This modification ensures that future readers can directly correlate the information in Fig. 1b with other sections of the manuscript, including Fig. 1c and the supplementary datasets. Importantly, this change solely impacts Fig. 1b and Supplementary Fig. 1.

Reviewer #1

The manuscript is well-written and provides a substantial amount of data. There are instances where the writing in the introduction can be improved. There are mixture of ideas presented within one paragraph.

- (1) Lines 68-73 - The statement describing the overall goal of the manuscript is not a typical component of the introduction and should be removed. The same is true for lines 91-94.

Response: We appreciate Reviewer #1's thoughtful requests. The statements that Reviewer #1 identified as misplaced in the introduction have been removed as requested.

Reviewer #2

The aim of this work was identifying clinically viable therapeutic strategies in TNBC and potential biomarkers of response. In particular, in this manuscript, the authors describe how PIM kinases are involved in the regulation of S100A8/A9 expression in TNBC and demonstrate the potential of targeting PIM in combination with anti-PD-1 antibodies to reduce tumor growth and increase cytotoxic immune responses in preclinical models. Overall, the rationale supporting the studies is well defined, the choices in the experimental design are properly justified, and the statistical analysis is complete. Thus, this study provides relevant and mechanistic knowledge on how S100A8/A9 regulates the immune component of the TME in vitro and preclinical models. Therefore, only a few aspects require further revision:

Major Comments:

- (1) Line 119: The prognostic potential of S100A8/A9 is evident with the background provided by the authors. However, the authors should clarify and elaborate on the use of S100A8/A9 as a predictive biomarker and specify the treatment for which it could be considered a predictive biomarker.

Response: We appreciate Reviewer #2's feedback. We acknowledge that our initial wording may have been misleading. Our intention was to convey that elevated S100A8/A9 expression could serve as a predictive biomarker of response to PIM kinase inhibitors. However, we agree that the way it was phrased could have been clearer. The concept of S100A8/A9 expression as a predictive biomarker of response to PIM kinase inhibitors is better articulated in the abstract and is discussed in detail in the discussion section. Therefore, we have removed the phrase “predictive biomarker of response” from the final paragraph in the introduction, as suggested.

- (2) To support the choice of the experimental cell model in Figure 2 (MDAMB468 cells), authors should consider including available data of the aggressiveness and response to treatments of the 4 TNBC cell lines analyzed (e.g., in preclinical models) and discuss the differences between the cell lines expressing S100A8/A9 and the ones that do not express S100A8/A9.

Response: We appreciate Reviewer #2's thoughtful comments. In this study, we utilized two established S100A8/A9+ human TNBC cell lines (MDA-MB-468 and BT-20) for in vitro mechanistic experiments (e.g., Fig. 2) and to obtain genetically defined TNBC cell supernatants (e.g., Fig. 5). We wish to clarify that our intention was not to suggest that the cell lines shown in Fig 2a themselves exhibit chemo-resistance or chemo-sensitivity. We have now explicitly

addressed this point in the methods section. As outlined in the discussion section, our emerging understanding in the field underscores the crucial role of robust cytotoxic immune responses in effectively killing TNBC/cancer cells within tumor tissues with chemotherapeutic agents. We agree with Reviewer #2's suggestion that it is essential to experimentally validate our working hypothesis, based on our bioinformatics analysis (Fig. 1a), that S100A8/A9+ TNBC tumors may demonstrate resistance to commonly used chemotherapeutic agents such as doxorubicin, cyclophosphamide, and paclitaxel using syngeneic mouse models of TNBC (e.g., Fig. 4b). Additionally, we share Reviewer #2's assessment regarding the need for a comprehensive understanding of the biological distinctions between S100A8/A9+ and S100A8/A9- cancer cells, beyond the few components currently identified, such as C/EBP β , STAT3, and PIM. We anticipate pursuing these avenues as independent projects in the future.

(3) Given the scope of Communications medicine and to support the potential use of S100A as a biomarker (that the authors highlight in the abstract, in the last paragraph of the introduction, as well as in the results section – Fig 4), some evidence should be provided in the context of breast cancer patient samples. In particular, as discussed by the authors, S100A8/A9 could be used as a serum biomarker. As the authors indicate, the correlation of serum S100A8/A9 levels with those of tumor samples should be determined in future studies. However, adding evidence of S100A8/A9 detection in blood samples of breast cancer patients and its association with the outcome will already strengthen the translational potential of their findings.

Response: We appreciate Reviewer #2's valuable suggestions. The newly added data (Fig. 5f) illustrate the determination of S100A8/A9 levels in serum samples from patients with TNBC at various breast cancer stages. This newly added data demonstrates the robust dynamic range of S100A8/A9 detection, with some samples showing serum levels exceeding 5.5 $\mu\text{g}/\text{mL}$, a threshold recently associated with poor responses to anti-PD-1 therapy in a melanoma study (ref 17). Notably, the newly added data reveals an intriguing finding – serum S100A8/A9 levels may not necessarily correspond to specific breast cancer stages. For example, samples from TNBC patients at Stage 1 exhibited S100A8/A9 levels ranging from “not detectable” (ND) to 1.23 $\mu\text{g}/\text{mL}$ and 5.73 $\mu\text{g}/\text{mL}$, while those at Stage 3 showed levels of 0.06 $\mu\text{g}/\text{mL}$ and 5.83 $\mu\text{g}/\text{mL}$. Our bioinformatics approach, as depicted in Fig 1a, suggests that TNBC patients with low serum S100A8/A9 levels (if they correspond to tumoral S100A8/A9 levels), regardless of their breast cancer stage (whether Stage 1 or Stage 3), might respond favorably to neoadjuvant chemotherapy. To test this hypothesis, our team is collaborating with clinical investigators here at Northwestern Memorial Hospital to initiate an IRB-controlled prospective study. We eagerly anticipate sharing the results of this study in the coming years.

Minor Comments:

(1) Line 56: Clarify that those findings were obtained in mouse models.

Response: Adjusted as requested.

(2) Line 77 and 167: Add reference.

Response: Adjusted as requested.

(3) Line 155: To provide a complete analysis, authors could add the Kaplan-Meier graphs for the other 2 cohorts not included in Figure S1B

(4) Line 161: HER2-positive tumors exhibit elevated S100A8 and A9 expression in all the cohorts, not in some.

Response: We would like to address Reviewer #2's Minor Comments #3 and #4 together as they are closely related.

The primary focus of this study has been on TNBC, with our bioinformatics approach (Fig. 1a) identifying elevated expression of S100A8 and -A9 in treatment-naïve Stage 1-3 TNBC tumors as indicative of subsequent disease progression following systemic treatments. We appreciate Reviewer #2 for bringing to our attention the need for clarification regarding the uniqueness of S100A8 and -A9 expression in the TNBC subset. As Reviewer #2 has astutely pointed out, our revised differential expression analysis reveals that S100A8 and -A9 exhibit significant and similar elevation not only in the TNBC subtype but also in the HER2 subtype (Supplementary Fig. 1d). This finding is noteworthy, suggesting that despite their differing receptor statuses, TNBC and HER2 subtypes may share treatment-resistant mechanisms, particularly in the context of the tumor microenvironment. While the scope of our study does not encompass the HER2 subtype, we hope that the presentation of this data in the main text and its depiction in the updated Supplementary Fig. 1 will encourage researchers focusing on the HER2 subtype to investigate the potential prognostic value and functional significance of S100A8/A9 expression in patients with HER2+ tumors.

(5) Line 259: Add “expression” before “levels.”

Response: Corrected as requested.

Reviewer #3:

In this manuscript, Begg et al. set out to unravel why some TNBC tumors progress despite existing therapies. To this end, they performed several bio-informatic meta-analyses on four publicly available datasets using semi-supervised approaches. The study identified 30 genes that were dysregulated in at least 3 of the 4 datasets, among which S100A9 came out as the most prognostic of the upregulated genes. A subsequent, less stringent analysis also identified the binding partner, S100A8 as a prognostic marker. The authors next performed a drug screening in a S100A8/A9-expressing TNBC cell line to identify compounds that modulate the expression of S100A8/A9. Most notably, PIM-inhibitors (PIM447, GDC-0339) consistently lowered the levels of heterodimeric S100A8/A9. This hit was subsequently validated using a series of convincing ELISA, Western blot and genetic CRISPR approaches, supporting that PIM is involved in the expression of S100A8/A9. Further mechanistic insight was gained by computationally inferring putative transcription factors that modulate S100A8/A9 expression (STAT3 and C/EBP β) and using a combination of knock-out experiments to evaluate how these are modulated by PIM. Finally, the authors use PDX mice models to demonstrate that PIM inhibition sensitizes S100A8/A9-high tumors to anti-PD1/PD-L1 antibody IC blockade, showing increased infiltration of cytotoxic T cells. Interestingly, the authors provide some evidence that S100A8/A9 levels in serum mirror the abundance in tumors, suggesting a potential application as a liquid biopsy biomarker.

Overall assessment of quality and importance:

This is a well-written, hypothesis-driven manuscript with rigorous methodology. There is indeed a lack of understanding how some TNBC tumors progress despite existing therapies, while others do not. This manuscript builds on previous knowledge that S100A8/A9 can induce an immunosuppressive TME in other cancer types. Importantly, this study also aimed to unravel the mechanisms behind this process and nominates PIM as an important player in S100A8/A9 expression. As such, this manuscript provides fundamental knowledge that is potentially applicable beyond the setting of TNBC. Although the applicability of S100A8/A9 in the serum as a biomarker needs further validation in a clinical context, as mentioned by the authors, this is an interesting aspect to the study indeed. Overall, I commend the authors for their work and support publication in Nature Communications Medicine. I only have a few minor comments listed below.

Minor Comments:

(1) The authors describe that S100A8/A9 can be expressed by several cell types, including cancer cells, cells of the myeloid lineage and TAMs. It is also indicated that in the murine models of TNBC, the major source of S100A8/A9 is cells of myeloid origin. As the study mainly focuses on the expression of S100A8/A9 in cancer cells, it would be informative to comment on the relative contribution of cancer cells to S100A8/A9 levels in the TME, compared to other cell types. Are these the major source of S100A8/A9 in the TME in a human context?

Response: We sincerely appreciate the insightful comments from Reviewer #3. This question raised by Reviewer #3 touches upon one of the fundamental and unresolved aspects of how S100A8/A9 may contribute to early and sustained tumor growth. Presently, we do not possess definitive answers to this question. However, we can offer one of the prevailing hypotheses in the field. As summarized in Bresnick et al. (ref 7), various tumor types have demonstrated the production of S100A8 and -A9, along with other members of the S100 protein family, by cancer cells. Similarly, non-cancer cells of myeloid lineage have been observed to produce S100A8 and -A9. These immunosuppressive myeloid cells are recruited to emerging tumor sites through the actions of pro-tumorigenic chemokines and cytokines, such as S100A8/A9, secreted by cancer cells. Once situated within the tumor microenvironment, these myeloid cells can further contribute to the pool of S100A8/A9 and other chemoattractants, perpetuating their recruitment. Interestingly, based on our preliminary observations (unpublished data), the production of S100A8/A9 by a single immunosuppressive human myeloid cell may exceed that of one human TNBC cell by a factor of more than ten. In one of the syngeneic mouse models used in this study, D2A1-M1, the balance between volume of cancer cells and non-cancer cells within tumor tissues varies with the stages of tumor development and growth. For instance, our ongoing, independent research reveals that the ratio between cancer cell counts and non-cancer cell counts in tumors at 50-75 mm³ can be as high as 5:1. However, this ratio may shift to 2:1 in tumors at 300 mm³, and eventually, it may become 1:1 in tumors exceeding 500 mm³. This variation underscores the critical roles non-cancer cells play in supporting continued tumor growth. Consequently, the dominant sources of S100A8/A9 are likely dependent on the developmental stages of cancer. To further support the significance of the data presented in Fig. 4d, we conducted an additional in vivo tumor growth study to provide a concrete example of S100A8 and -A9 expression in cancer and non-cancer cells within size-controlled D2A1-M1 tumors (at 500 mm³) (new Supplementary Fig. 4d).

(2) I support the idea of a consensus approach to identify prognostic markers. However, I would have liked to see a summary table listing the main characteristics of the four datasets used in the study, in order to evaluate the variance among them. Especially parameters like sample size, gender distribution, age ranges would be useful to compare the different studies at a glance.

Response: We are grateful for Reviewer #3's valuable suggestion. We agree with Reviewer #3 that summarizing and presenting the clinical parameters in a table format would enhance the informativeness of our manuscript. However, achieving this in a well-organized and presentable manner presents certain challenges. Firstly, it is important to note that our study primarily focuses on the TNBC subset. Our unique bioinformatics approach, as illustrated in Fig. 1a, supplementary Fig. 1a, and detailed in the Methods, was exclusively applied to TNBC samples within each dataset, representing only a portion of the overall dataset used in our study. While two of the original studies did provide patient characteristics in a tabular format, this information encompasses all breast cancer subtypes and lacks specificity to TNBC patients. Secondly, each study employed different parameters for present patient characteristics, including variations in mean vs. median ages and different age cut-offs or age range representations. Lastly, one of the most widely-used datasets, the Yau dataset (also described as a "chemotherapy naïve dataset" in the literature), combines data from four independent datasets and does not include a summary table displaying patient ages. For readers seeking detailed patient characteristics, we have included ample information, including citations and references to publicly accessible web-based databases, in the Methods section. Supplementary Fig. 1b provides details on sample sizes and breast cancer clinical subtypes.

(3) Did you start from processed count tables or raw fastq files of the four datasets? If starting from count tables, some comment on the influence of the pre-processing steps (QC, mapping, etc.), which probably differed among the studies, on the final data would be in place.

Response: We appreciate Reviewer #3's question. It is indeed a crucial consideration that the choice of expression platforms used for profiling and data pre-processing can significantly impact the final data and potentially influence findings. In fact, this recognition underlies the rationale for our bioinformatics workflow, which initially assessed each dataset independently. Genes that consistently exhibited associations with subsequent disease progression across multiple datasets, even in the presence of variations in how the expression data was generated or processed, as well as differences in other underlying clinical characteristics, were deemed more robust. Such genes were accorded higher priority and selected for further comprehensive evaluation.

(4) Regarding the group of 30 genes that were most robustly associated with disease progression: are the genes with the highest significance scores (GATA3, NAT1, ...) known to be involved in TNBC disease progression? If so, this would validate the approach, giving more confidence to the S100A9 hit. Please elaborate.

Response: We appreciate this intriguing question from Reviewer #3. Indeed, elevated expression of GATA3 and NAT1 has been independently and strongly associated with improved prognosis. However, it is worth noting that these genes have been exclusively studied within the context of hormone-receptor-positive breast cancer, and much is unknown regarding their function in TNBC. To address this important point, we have now incorporated additional sentences in the Results section (corresponding to Fig. 1) to highlight these observations.

(5) Regarding the gene-wise bootstrap scores used for sample stratification (lines 421-423), it would be useful to provide the readers with the ranked list for future reference.

Response: We have now included the requested ranked lists in Supplementary Data 1, along with the final gene set.

(6) On the same topic, I would be interested to see which genes are in the optimal multi-gene classifier (lines 431-432), which was used to stratify the samples. How many genes were used in the end? Which genes? Are S100A8/A9 or other hit genes part of the classifier?

Response: The lists of optimal classifiers for each dataset have been included in Supplementary Data 1, along with the final gene set and the intermediate products mentioned above. The classifiers are composed of the genes that ultimately became the final hits. However, the classifiers happen to not include S100A8 or -A9.

(7) Some TNBC cell lines are high in S100A8/A9, while others do not seem to express this. Is anything known about how these behave differently in PDX models? One would expect the S100A8/A9-low cell lines to progress slower compared to the S100A8/A9-high cell lines. Please comment.

Response: We appreciate Reviewer #3's questions. Based on the data presented in Fig. 4b, e, f, and g, we find that tumor growth rates of the syngeneic lines utilized in this study do exhibit some degree of proportionality to tumoral S100A8 and -A9 levels, as Reviewer #3 noted. Nevertheless, we have not conducted functional experiments to conclusively demonstrate that the observed differences in growth kinetics can be attributed to variances in tumoral S100A8 and -A9 expression within the models used in this study. It is important to note that previous reports have shown that tumors from various origins experienced significantly slower growth in mice when lacking S100A8 and -A9, as described in the introduction. Considering our working hypothesis, depicted in Fig. 1, we anticipate that TNBC tumors exhibiting elevated S100A8 and -A9 expression may display reduced sensitivity to certain chemotherapeutic agents such as doxorubicin, cyclophosphamide, and paclitaxel. We plan to experimentally test this idea in an independent study.

REVIEWERS' COMMENTS:

Reviewer #2 (Remarks to the Author):

The authors have addressed all my queries satisfactorily and I am happy to recommend acceptance of the revised manuscript.

Reviewer #3 (Remarks to the Author):

The authors perfectly addressed my remarks, I have no additional comments. I recommend publication in the current form.